# On the discrepancy of HCl processing in the core of the wintertime polar vortices

Jens-Uwe Grooß[1], Rolf Müller[1], Reinhold Spang[1], Ines Tritscher[1], Tobias Wegner[1,2],
Martyn P. Chipperfield[3], Wuhu Feng[3,4], Douglas E. Kinnison[5], and Sasha Madronich[5]

[1]Institut für Energie- und Klimaforschung – Stratosphäre (IEK-7), Forschungszentrum Jülich, Germany
[2]now at: KfW Bankengruppe, Frankfurt, Germany
[3]School of Earth and Environment, University of Leeds, UK
[4]National Centre for Atmospheric Science, University of Leeds, Leeds, UK
[5]Atmospheric Chemistry Observations and Modeling Laboratory, National Center for Atmospheric Research, Boulder, USA

**Correspondence:** Jens-Uwe Grooß (j.-u.grooss@fz-juelich.de)

**Abstract.**

More than three decades after the discovery of the ozone hole, the processes involved in its formation are believed to be understood in great detail. Current state-of-the-art models can reproduce the observed chemical composition in the springtime polar stratosphere, especially regarding the quantification of halogen-catalysed ozone loss. However, we report here on a discrepancy between simulations and observations during the less-well studied period of the onset of chlorine activation. During this period, which in the Antarctic is between May and July, model simulations significantly overestimate HCl, one of the key chemical species, inside the polar vortex during polar night. This HCl discrepancy is also observed in the Arctic. The discrepancy exists in different models to varying extents; here we discuss three independent ones, the Lagrangian model CLaMS as well as the Eulerian models SD-WACCM and TOMCAT/SLIMCAT. The HCl discrepancy points to some unknown process in the formulation of stratospheric chemistry that is currently not represented in the models.

We characterise the HCl discrepancy in space and time for the Lagrangian chemistry-transport model CLaMS, in which HCl in the polar vortex core stays about constant from June to August in the Antarctic, while the observations indicate a continuous HCl decrease over this period. The somewhat smaller discrepancies in the Eulerian models SD-WACCM and TOMCAT/SLIMCAT are also presented. Numerical diffusion in the transport scheme of the Eulerian models is identified to be a likely cause for the inter-model differences. Although the missing process has not yet been identified, we investigate different hypotheses on the basis of the characteristics of the discrepancy. An underestimated HCl uptake into the PSC particles that consist mainly of $H_2O$ and $HNO_3$ cannot explain it due to the temperature correlation of the discrepancy. Also, a direct photolysis of particulate $HNO_3$ does not resolve the discrepancy since it would also cause changes in chlorine chemistry in late winter which are not observed. The ionisation caused by Galactic Cosmic Rays provides an additional $NO_x$ and $HO_x$ source that can explain only about 20% of the discrepancy. However, the model simulations show that a hypothetical decomposition of particulate $HNO_3$ by some other process not dependent on the solar elevation, e.g. involving Galactic Cosmic Rays, may be a possible mechanism to resolve the HCl discrepancy. Since the discrepancy reported here occurs during the beginning of the

chlorine activation period, where the ozone loss rates are small, there is only a minor impact of about 2% on the overall ozone column loss over the course of Antarctic winter and spring.

## 1   Introduction

The discovery of large ozone depletion in the Antarctic spring, now known as the ozone hole (Farman et al., 1985; Stolarski
et al., 1986), came as a complete surprise, although the effect of chlorine-catalysed ozone depletion in the stratosphere had been discovered since the 1970s (Molina and Rowland, 1974). Nevertheless, after its discovery, the processes leading to the formation of the ozone hole were rapidly established (Solomon et al., 1986; Crutzen and Arnold, 1986; Anderson et al., 1989) and are now understood in great detail (e.g. Solomon, 1999). Briefly, (1) the decomposition of anthropogenically emitted source gases, such as chlorofluorocarbons (CFCs), releases chlorine in the stratosphere, which is first converted into the passive non-ozone-
depleting reservoir compounds HCl and $ClONO_2$. (2) Polar stratospheric clouds (PSCs) form at the very low temperatures in the polar winter and spring. These are liquid or crystalline particles that mainly consist of condensed $HNO_3$ and $H_2O$ taken up from the gas phase. PSC particles, as well as the cold stratospheric sulphate aerosol, provide surfaces on which heterogeneous chemical reactions can occur. (3) The chlorine reservoir species HCl and $ClONO_2$ are activated into $Cl_2$ and HOCl by heterogeneous reactions on the PSCs and on the cold aerosols. The heterogeneous reaction of HCl with $ClONO_2$
plays a major role in this chlorine activation process (Solomon, 1990; Jaeglé et al., 1997). (4) In the presence of sunlight in polar spring, ozone is depleted by catalytic cycles involving the active chlorine (Molina and Molina, 1987) and bromine compounds. (5) Ozone-depleted air-masses are trapped within the polar vortex due to the efficient dynamical transport barrier at the polar vortex edge (Schoeberl et al., 1992).

Since the first theoretical explanation of the ozone hole, many research activities have focused on the detailed processes
and on quantitative numerical simulations of the ozone depletion mechanisms. Research activities include studies of gas-phase kinetics, heterogeneous chemistry, catalytic ozone loss cycles, liquid and solid PSC formation, PSC sedimentation and transport barriers at the polar vortex edge. Today it is known that there is a variety of particles present at low temperatures in the polar vortices (e.g. Pitts et al., 2011; Spang et al., 2018; Pitts et al., 2018), namely ice particles of different sizes and number densities, nitric acid trihydrate particles (NAT), liquid super-cooled ternary $H_2SO_4/H_2O/HNO_3$ solution particles (STS) and cold binary
sulphate aerosol. All of these particles can, in principle, facilitate the heterogeneous reactions that lead to activation of chlorine in the polar stratosphere (Peter, 1997; Solomon, 1999; Wegner et al., 2012; Drdla and Müller, 2012; Shi et al., 2001; Grooß et al., 2011). An important impact of the PSCs is the uptake of $HNO_3$ into the particles which deplete the gas-phase of $NO_x$ as long as the particles exist (Crutzen and Arnold, 1986; Müller et al., 2018). Particles such as NAT, present in low number densities and in turn large sizes, are also able to transport downward significant amounts of reactive nitrogen ($NO_y$), resulting
in denitrification of the stratospheric air (Fahey et al., 2001; Grooß et al., 2005; Molleker et al., 2014).

These processes are included in state-of-the-art models which can reproduce the observed polar ozone depletion. Therefore, WMO (2014) states that "well understood stratospheric chemical processes involving chlorine and bromine are the primary cause of the seasonal polar ozone depletion". However, the phase of initial chlorine activation in the polar vortices has not yet

been studied in similar detail. During this phase there is a difference in HCl between simulations and observations, which was first recognised by Wegner (2013). Here we report on this observation-simulation discrepancy for HCl in the polar stratosphere during the period of initial chlorine activation in the core of the polar vortex. In the polar vortex stratosphere, chlorine activation by heterogeneous reactions starts in early winter and can be indirectly detected by a depletion of the observed HCl mixing ratios. Wegner (2013) showed that in the dark core of the polar vortex, the observed depletion of HCl is much faster than in simulations of the specified dynamics version of the Whole Atmosphere Community Climate Model (SD-WACCM).

This discrepancy, hereafter referred to as the "HCl discrepancy", has been shown in other publications, although it was mostly not the focus of those studies. Brakebusch et al. (2013) show the HCl discrepancy in a simulation for the Arctic winter 2004/2005. It could be partly corrected in the vortex average by decreasing the temperature in the module for heterogeneous chemistry by 1 K. Solomon et al. (2015) also show the discrepancy in SD-WACCM for early winter 2011. It is present at $82°S$ and $53\,hPa$ (their Fig. 4) and at $80°S$ and $30\,hPa$ (their Fig. 8) in all of the sensitivity studies shown. However, the focus of that paper was on the late winter and spring period and the issue was not discussed further. Kuttippurath et al. (2015) show 10 years of simulation with the model MIMOSA-CHIM. They compare the time dependence of vortex average mixing ratios with MLS observations and present an average of the 10 Antarctic winters 2004-2013. In their Fig. 4, the discrepancy also seems to be present, even though it is smoothed out by the averaging procedure. Recently Wohltmann et al. (2017) did explicitly address the HCl discrepancy. They show simulations with the Lagrangian model ATLAS in comparison with MLS observations for the winter Arctic 2004/2005. Although the comparison with the other chemical compounds is very good, for example with MLS $N_2O$ and ozone, the observed HCl mixing ratios also indicate a depletion in early winter that is not present in the vortex-average simulation shown. As a possible solution they suggest an increased uptake into the liquid STS particles due to a higher solubility by imposing an artificial negative temperature offset of 5 K. With that, the vortex average HCl mixing ratios decreased more in early winter. However, there is no evidence for what the reason of this enhanced solubility could be. Santee et al. (2008) also show a MLS-model comparison of HCl with apparently the opposite problem, that is a modelled depletion of HCl before it was observed and with a larger vertical extent. These simulations were performed by an earlier version of the TOMCAT/SLIMCAT model and a simple PSC scheme that, for example, triggers PSC formation directly at the NAT equilibrium temperature. Further, the initial $ClONO_2$ in this study could not be constrained with observations. We therefore concentrate here on the simulations with the updated version of TOMCAT/SLIMCAT.

The HCl discrepancy documented here is investigated in detail by employing several well-established models, (1) the Lagrangian chemistry transport model CLaMS (Grooß et al., 2014), (2) SD-WACCM (Marsh et al., 2013) and (3) TOMCAT/SLIMCAT (Chipperfield, 2006). The models CLaMS, SD-WACCM and TOMCAT/SLIMCAT are described briefly in Section 2. The observations used are satellite data from the Microwave Limb Sounder (MLS) (Froidevaux et al., 2008) and the Michelson Interferometer for Passive Atmospheric Sounding (MIPAS) (Höpfner et al., 2007), which are described in Section 3. The model results and comparison with the observations are shown in Section 4. The discussion in Sections 5 and 6 shows the characteristics of the likely missing process in the models and discusses some hypotheses about which process could resolve the HCl discrepancy. In Section 7, we quantify the extent to which a potential additional chlorine activation mechanism would impact polar ozone loss.

## 2 Model descriptions

We present simulations from the three models CLaMS, WACCM and TOMCAT/SLIMCAT for the Antarctic winter 2011. This year had a rather large ozone hole with a size within the top ten of the last 4 decades (Klekociuk et al., 2014) and was chosen because both MIPAS and MLS data were available. The setup of the models with respect to initialisation, boundary conditions, transport schemes and PSC formation, are different which also contributes to differences between the model results. The formulation of stratospheric chemistry, especially heterogeneous chemistry of chlorine compounds is implemented as commonly used (Jaeglé et al., 1997) and recommended and is comparable among the three models.

### 2.1 CLaMS

The Chemical Lagrangian Model of the Stratosphere (CLaMS) is the main tool used here for the standard simulations and sensitivity tests. CLaMS is a Lagrangian chemistry-transport model that is described elsewhere (McKenna et al., 2002a, b; Grooß et al., 2014, and references therein). The model grid points are air parcels that follow trajectories and are therefore distributed irregularly in space. Mixing between the air parcels is calculated by an adaptive grid algorithm that depends on the large-scale horizontal flow deformation (McKenna et al., 2002a). The change of composition by chemistry and especially heterogeneous chemistry is calculated along the trajectories (McKenna et al., 2002b; Grooß et al., 2014). Recent developments include the replacement of the previous chemistry solver routine IMPACT by a Newton-Raphson method derived from Wild and Prather (2000) as described by Morgenstern et al. (2009). PSC particles are simulated along individual trajectories including their gravitational settling. While in the study by Grooß et al. (2014) the Lagrangian PSC particle sedimentation scheme only simulated NAT particles, we include here an update that also simulates ice particles as described by Tritscher et al. (2018). With this parameterisation the model is able to sifficiently reproduce the observed PSC types and distribution.

The model setup is very similar to that of Grooß et al. (2014). Initialisation and boundary conditions are derived from MLS observations of $O_3$, $N_2O$, $H_2O$, and HCl. The tropospheric domain is derived from a multi-annual simulation of CLaMS (Pommrich et al., 2014). Specific tracer-tracer correlations are used to derive the remaining chemical species. The simulation has 32 vertical levels below 900 K potential temperature with a vertical resolution of about 900 m in the lower stratosphere and a horizontal resolution of 100 km. The simulation extends from 1 May 2011 until 31 October 2011. For comparison we also show simulations with the same setup for the Arctic winter 2015/2016 starting at 1 November 2015 until 31 March 2016. However, in this paper we focus mainly on the Antarctic case.

### 2.2 WACCM

The Community Earth System Model version 1 (CESM1), Whole Atmosphere Community Climate Model (WACCM), is a coupled chemistry climate model from the Earth's surface to the lower thermosphere (Garcia et al., 2007; Kinnison et al., 2007; Marsh et al., 2013). WACCM is superset of the Community Atmosphere Model, version 4 (CAM4), and includes all of the physical parameterisations of CAM4 (Neale et al., 2013) and a finite volume dynamical core (Lin, 2004) for the tracer advection. The horizontal resolution is 1.9° latitude × 2.5° longitude and has 88 vertical levels up to about 150 km. The ver-

tical resolution in the lower stratosphere ranges from 1.2 km near the tropopause to about 2 km near the stratopause; in the mesosphere and thermosphere the vertical resolution is about 3 km. For this work, the specified dynamics (SD) option is used (Lamarque et al., 2012). Here, temperature, zonal and meridional winds, and surface pressure are used to drive the physical parameterisation that control boundary layer exchanges, advective and convective transport, and the hydrological cycle.

The meteorological analyses are taken from the National Aeronautics and Space Administration (NASA) Global Modeling and Assimilation Office (GMAO) Modern-Era Retrospective Analysis for Research and Applications (MERRA) (Rienecker et al., 2011) and the nudging approach is described by Kunz et al. (2011). The simulation used in the work is based on the International Global Atmospheric Chemistry / Stratosphere-troposphere Processes And their role in Climate (IGAC/SPARC) Chemistry Climate Model Initiative (CCMI) (Morgenstern et al., 2017). CESM1 (WACCM) includes a detailed representation

of the chemical and physical processes in the troposphere through the lower thermosphere. The species included within this mechanism are contained within the $O_x$, $NO_x$, $HO_x$, $ClO_x$, and $BrO_x$ chemical families, along with $CH_4$ and its degradation products. In addition, 20 primary non-methane hydrocarbons and related oxygenated organic compounds are represented along with their surface emissions. There is a total of 183 species and 472 chemical reactions; this includes 17 heterogeneous reactions on multiple aerosol types (i.e., sulphate, nitric acid trihydrate, and water-ice). Details on the stratospheric heterogeneous

chemistry can be found in Wegner et al. (2013) and Solomon et al. (2015).

The treatment of PSCs follows the methodology discussed in Considine et al. (2000) and is described by Kinnison et al. (2007) and Wegner et al. (2013). Surface area densities of the sulphate binary background aerosol is taken from the Chemistry Climate Model Initiative (CCMI) recommendation (Arfeuille et al., 2013). STS formation is calculated by the WACCM Aerosol Physical Chemistry Model (APCM) (Tabazadeh et al., 1994). NAT formation is suggested to occur at a supersaturation of 10,

about 3 K below the thermodynamic equilibrium temperature $T_{NAT}$. The surface area density and radius for STS and NAT assume a log-normal size distribution with a width of 1.6. The particle number concentration is set to $10\,cm^{-3}$ and $0.01\,cm^{-3}$ for STS and NAT, respectively.

## 2.3 TOMCAT/SLIMCAT

TOMCAT/SLIMCAT (Chipperfield, 2006) is a gridpoint, Eulerian off-line 3-D chemical transport model which has been
widely used for simulations of stratospheric chemistry (e.g. Dhomse et al., 2016; Chipperfield et al., 2017). Tracer transport is performed using the conservation of second-order moments scheme of Prather (1986). The model has a detailed description of stratospheric chemistry including reactions of the oxygen, nitrogen, hydrogen, chlorine and bromine families. Heterogeneous chemistry is treated on sulphate aerosols as well as liquid and solid PSCs.

Surface area densities of the sulphate binary background aerosol are read in from monthly mean fields obtained from
30 ftp://iacftp.ethz.ch/pub_read/luo/CMIP6/ (Arfeuille et al., 2013; Dhomse et al., 2015). These are converted to $H_2SO_4$ volume mixing ratio which is a passive model tracer using the model local temperature and $H_2O$ fields. At each model gridpoint and timestep the $H_2SO_4$ is converted to STS using the composition expression of Carslaw et al. (1995). HCl is modelled to be taken up in STS using the expression of Luo et al. (1995). The size distribution of liquid sulphate aerosol is calculated assuming a log-normal size distribution with width of 1.8 and 10 drops per $cm^3$. The model version used here has a simple equilibrium

scheme for the formation of nitric acid trihydrate particles, with an assumed supersaturation factor of 10 (Feng et al., 2011). The model considers NAT particles of both small ($1\,\mu$m) and large ($10\,\mu$m) diameter. The condensed NAT particles are first assumed to form small particles with $1\,\mu$m diameter and a numbers density of $1\,\text{cm}^{-3}$. Any excess condensed $HNO_3$ is assumed to form large $10\,\mu$m particles. Denitrification occurs through sedimentation of both types of NAT particles with appropriate fall
speeds.

The model has a variable horizontal and vertical resolution. The model longitudes are regularly spaced while the latitude spacing can vary. Typically, the model is run using standard Gaussian grids associated with certain spectral resolutions. For the simulations presented here the model used either a $2.8° \times 2.8°$ resolution (T42 Gaussian grid) or a $1.2° \times 1.2°$ (T106 Gaussian grid) resolution. When forced by ECMWF analyses (as here) the model reads 6-hourly analyses of temperature, vorticity,
divergence, humidity and surface pressure as spectral coefficients. These quantities are averaged onto the model grid by spectral transforms. For both resolution experiments used here, ECMWF data at T42 resolution were used. The 6-hourly gridpoint meteorological fields are interpolated linearly in time. Near the pole the model groups gridboxes for transport in the east-west direction so that the tracer transport remains stable, which effectively degrades the resolution somewhat. In the runs used here vertical motion was calculated from the large-scale mass flux divergence on the model grid. The model used 32 hybrid
sigma-pressure levels from the surface to about $60\,\text{km}$, with a resolution of 1.5-2.0 km in the lower stratosphere.

For the runs presented here the low resolution ($2.8° \times 2.8°$) simulation was initialised in 1977 and fully spun-up for the simulation of 2011. From 1989 the simulation was forced by ERA-Interim reanalyses. For the $1.2° \times 1.2°$ resolution simulation output from the low resolution simulation was interpolated to the higher resolution grid on January 1st, 2011 and the model integrated for 1 year. The use of pressure levels in the lower stratosphere can enhance the effects of numerical diffusion
compared to theta levels (Chipperfield et al., 1997).

## 3   Data description

### 3.1   MLS data

The HCl observations by the Microwave Limb Sounder (MLS) on board the Aura satellite are the main data set used in this study (Froidevaux et al., 2008). MLS observes in limb viewing geometry on the so-called A-train orbit, circling the earth
15 times daily covering latitudes from 82°S to 82°N. We use the MLS version 4.2 data (Livesey et al., 2017). In the lower stratospherte the vertical resolution is about $3\,\text{km}$ and the accuracy of HCl observations is about 0.2 ppbv. MLS also observes ClO mixing ratios with an accuracy in the lower stratosphere of about 0.2 ppbv. The MLS observations of $O_3$, $N_2O$, and $H_2O$ have also been used to constrain the CLaMS model initialisation and boundary conditions.

### 3.2   MIPAS data

In addition to MLS, we also use data from the Michelson Interferometer for Passive Atmospheric Sounding (MIPAS) on Envisat. MIPAS operated between 2002 and 2012 and observed in limb geometry on about 15 orbits per day spanning the

latitude range from 87°S to 89°N. In particular, we use ClONO$_2$ mixing ratios employing the KIT retrieval in the version V5R_CLONO2_221. The vertical resolution in the lower stratosphere is about 3-4 km and the accuracy of the ClONO$_2$ observations, derived from correlative measurements, is about 0.05 ppbv (Höpfner et al., 2007). However, it is difficult to retrieve gas-phase mixing ratios from the infrared spectra in the presence of PSCs due to optical interference. For the frequently optical thick PSC spectra observed by MIPAS, a retrieval of gas-phase mixing ratios is likely impossible. Therefore, data gaps due to PSCs are often present in the cold Antarctic stratosphere.

## 3.3 PSC detection by MIPAS and CALIOP

We also exploit the ability of satellite measurements to detect PSCs. We use the retrieval of PSCs from the infrared spectra of the Envisat MIPAS experiment (Spang et al., 2016). We also use the detection of PSCs from space-borne Cloud-Aerosol Lidar with Orthogonal Polarization (CALIOP) on the CALIPSO satellite (Pitts et al., 2013, 2018). CALIOP observes from 15 orbits per day reaching latitudes of up to 82°in both hemispheres. It is possible to distinguish between the characteristics of different PSC types (STS, NAT, ice) from both instruments. However, here we do not discriminate between different PSC types, since they often occur coincidently and the PSC type that dominates the optical signal may not be the relevant type here.

## 4 Development of HCl in polar stratospheric winter

The development of HCl in polar stratospheric winter is strongly influenced by heterogeneous chlorine activation. The heterogeneous reactions that cause this typically occur below a temperature threshold of about 195 K on cold binary aerosol or on PSCs composed of either super-cooled liquid ternary HNO$_3$/H$_2$SO$_4$/H$_2$O solution (STS), water ice or crystalline nitric acid trihydrate (NAT). An important and fast HCl sink is the heterogeneous reaction

$$ClONO_2 + HCl \rightarrow HNO_3 + Cl_2$$

(Solomon et al., 1986). At the beginning of winter, as temperatures drop below the threshold for heterogeneous reactions to occur, this reaction quickly depletes HCl typically until all available ClONO$_2$ has reacted, since HCl is usually available in large excess (Jaeglé et al., 1997). This step is therefore referred to as the "titration step". After that, since no reaction partner is available, HCl mixing ratios would remain constant in the dark in the absence of transport or mixing processes. A further depletion of HCl is only possible if there is a reaction partner. This reaction partner first needs to be formed, and in addition to ClONO$_2$ it could be HOCl which would then react by the heterogeneous reaction

$$HOCl + HCl \rightarrow H_2O + Cl_2$$

(Solomon, 1999). The resulting geographic distribution HCl and its development over the winter is presented in Fig. 1. The figure shows maps of MLS HCl observations (Froidevaux et al., 2008) and corresponding CLaMS results on the 500 K potential temperature level for selected days in May, June and July. On 21 May, when the HCl depletion due to heterogeneous processing is first apparent, the model seems to represent the observations well. By 20 June, HCl depletion has mainly occurred at the vortex edge, where it is more distinctly pronounced in the observations. The comparison on 20 July clearly demonstrates the

discrepancy between simulations and observations. The model shows HCl mixing ratios well above 1.5 ppbv in the vortex core, while the MLS observations indicate almost zero values of HCl, within the limits of the total estimated uncertainty of the data.

To investigate the HCl discrepancy further, we averaged both the observations and simulation for each day with respect to equivalent latitude (Butchart and Remsberg, 1986) and potential temperature using 20 equivalent latitude ($\Phi_e$) bins of equal area between $\Phi_e$=50°S and 90°S. Figure 2 shows the characteristics of the observed HCl development by displaying cuts through the equivalent latitude/potential temperature space. The top left panel shows the time development of the vortex-core average ($\Phi_e$ >75°S) as a function of potential temperature. The top right panel shows the time development on the 500 K potential temperature level. Grey lines correspond to the position of the cuts or borders displayed in the other panels. The bottom two panels show the $\Phi_e/\theta$ cross-sections for two selected days, 20 June and 20 July. Figure 3 shows the corresponding results from the CLaMS simulation. The top panels of Figs. 2 and 3 agree well with respect to the time at which the HCl depletion starts at the different altitudes and latitudes. However, it appears that after the first "titration" step, HCl depletes much more slowly in the simulations compared to the MLS observations. From the comparison of Figs. 2 and 3, it is evident that the model HCl discrepancy is present over a wide altitude range throughout polar winter.

Figure 4 shows the time-series of the similarly averaged HCl mixing ratio for the Arctic winter 2015/2016, both for the MLS observations (top panels) and the CLaMS simulation (bottom panels). This winter was chosen because it was particularly cold in the stratosphere (Dörnbrack et al., 2017). Here, the HCl discrepancy is also apparent, although to a lesser extent than in the Antarctic. It is most evident at the slightly lower potential temperature of about 475 K in the Arctic vortex core. Compared with the observations, the model requires over one month longer, until February, to reach the minimum HCl mixing ratios. While the discrepnacy is therefore also present in the Arctic, in the following we focus on the Antarctic results where it is much more strongly pronounced.

For further comparison, we now show results for the Antarctic winter 2011 simulated by the Eulerian models SD-WACCM and TOMCAT/SLIMCAT. Figure 5 shows the simulated HCl mixing ratios from these models plotted identically as in Fig. 1. The HCl discrepancy is clearly also evident in these simulationsi, as on 20 July, by when HCl had disappeared in the MLS data, both models also show an area of elevated HCl still present in the polar vortex core, although with lower HCl mixing ratios compared to CLaMS. The HCl depletion in the vortex core is stronger in the TOMCAT/SLIMCAT simulation than in SD-WACCM, but still not comparable to the MLS observations. For a more detailed comparison, the results of SD-WACCM and TOMCAT/SLIMCAT have been averaged in $\Phi_e/\theta$ space in a similar manner to CLaMS and the observations. The WACCM model daily output interpolated to potential temperature levels was averaged using potential vorticity and equivalent latitude calculated from ERA-Interim data. The TOMCAT/SLIMCAT model output was directly averaged within the internal calculated equivalent latitude, also based on ERA-Interim data. A depiction corresponding to Fig. 3 but for SD-WACCM and TOMCAT/SLIMCAT is shown as Figs. S1-S3 in the Supplementary Material. Figure 6 shows a time-series of the vortex-core-average HCl and ClONO$_2$ simulated by the three models. The green line corresponds to the MLS and MIPAS observations for HCl and ClONO$_2$, respectively. However, only very limited ClONO$_2$ observations are available from June to August, since the presence of PSCs impede the MIPAS retrieval process. There are differences between the individual simulations that arise due to different model formulation and initialisation and that are discussed below. Generally, the Eulerian models SD-WACCM

and TOMCAT/SLIMCAT show a faster HCl depletion than CLaMS. However, it seems that all models underestimate the rate of HCl depletion after the titration step.

The three simulations presented show that the initial titration step is completed by the end of May and no further strong changes of HCl occur until early July. Differences during May are likely caused by different initial chlorine partitioning. In July and August, HCl decreases further, CLaMS being the model that indicates the slowest HCl decrease. There is even an apparent HCl increase in CLaMS, that is caused by the descent of airmasses with higher inorganic chlorine $Cl_y$.

One possible reason for the inter-model difference could be differences in the chemistry formulation of the different models. To investigate this hypothesis, box model versions of TOMCAT/SLIMCAT and CLaMS were integrated along an example trajectory. This trajectory was chosen from an ensemble of 1800 vortex trajectories of 2.5-month duration, along which the ozone and HCl development was closest to the vortex core average shown in Fig. 6. This trajectory should therefore be representative for the vortex core average ozone and HCl development in the 3-D model. For this example trajectory, no significant differences were found between the models. Both models similarly showed no further depletion of HCl after the titration step in the dark unmixed polar night conditions (see Fig. S4 of the Supplementary Material). Thus, the differences among the models are unlikely due to model differences in the formulation of the chemical scheme.

A large part of the the inter-model difference may be caused by the fact that the Lagrangian formulation of the model CLaMS minimises numerical diffusion while it is inherent in the Eulerian models SD-WACCM and TOMCAT/SLIMCAT. The numerical diffusion arises in Eulerian models because on each time-step the chemical tracers are typically averaged within the extent of a model grid box. The minimisation of the numerical diffusion from this lower resolution limit depends on the performance of the advection scheme, for example the Prather (1986) scheme used in TOMCAT/SLIMCAT stores sub-gridscale distributions of the tracers. In practice, all Eulerian models likely aim to have tracer advection schemes which limit numerical diffusion, although it will always occur to some extent. The artificial mixing by numerical diffusion could provide additional $ClONO_2$ or $HOCl$ as a reaction partner for HCl or $NO_x$ that would form $ClONO_2$. To investigate this hypothesis, a sensitivity simulation by CLaMS was performed, in which a regular Eulerian grid with $2.8° \times 2.8°$ horizontal spacing and the vertical model grid (with the vertical resolution of 800-1000 m in the lower stratosphere) as vertical levels were defined. Every 24 h of the model run it was checked, whether more than one air parcel resides in the same grid box. These air parcels were then set to the average mixing ratio of all air parcels residing in the grid box. Although an Eulerian model that must average over these grid boxes on every time-step would be even more diffusive, this study can at least show a lower limit of this effect. The corresponding HCl result (labelled "mix-Euler") is shown as orange line in Fig. 6. Indeed, the additional imposed Eulerian mixing causes a faster HCl depletion until mid-August. Although it is difficult to mimic additional numerical diffusion in a Lagrangian model, this sensitivity supports the assumption that at least part of the difference between CLaMS and the other Eulerian models maybe caused by numerical diffusion. Thus it is likely, that the numerical diffusion in the Eulerian models masks part of the effect responsible for the HCl discrepancy investigated in this study. However, it should be noted there is not much difference in the development of HCl between the two resolutions of TOMCAT/SLIMCAT. In the low resolution simulation, the gradients of ozone and other trace species, especially at the vortex edge are weaker, but the time series of

vortex-core averages are very similar. Potentially, in the high resolution simulation the numerical diffusion may already be significant.

In the following, we list potential model deficiencies that could be responsible for the HCl discrepancy. (1) If the initial $ClONO_2$/HCl ratio was not set to realistic values, this would have immediate impacts on the first titration step of the reaction HCl+$ClONO_2$. This should have been avoided at least in the CLaMS simulations as both $ClONO_2$ and HCl have been initialised from observations. (2) An error with respect to the transport though the vortex edge or to mixing within the vortex could cause false results. This may affect the results in a similar way to the effect of numerical diffusion discussed above. However, this would likely rather cause a discrepancy near the vortex edge but not in the vortex core. (3) An increased uptake of HCl into PSC particles potentially caused by an underestimation of the solubility of HCl. As mentioned above, Wohltmann et al. (2017) suggest an empirical correction to resolve the discrepancy in which they apply a -5 K bias within the calculation of the HCl uptake into liquid particles. This correction is able to decrease the HCl discrepancy in the vortex average and is investigated below. However, a significant vertical transport of HCl by particle sedimentation similar to that of $HNO_3$ should cause a significant depletion of total chlorine ($Cl_y$) at the end of the polar winter which is not observed. (4) There could be unknown heterogeneous chemical reactions. Wohltmann et al. (2017) briefly discuss an additional heterogeneous reaction involving HCl and conclude that this cannot be excluded. However, it must fulfil the conditions that it does not change the remaining chemical composition to a large extent. (5) A temperature bias of the underlying meteorological analyses would in principle be a possible factor. However, during the period of initial chlorine activation it would likely not have a significant impact, as the HCl depletion is limited by the availability of $ClONO_2$, not by the strongly temperature-dependent heterogeneous reaction rate. (6) An offset in the water vapour mixing ratio. The formation of PSCs and uptake into PSCs depends on the amount of gas phase water vapour in the model. Therefore, it is also important to verify that the models do not have a significant offset in water vapour mixing ratio. In the case of the CLaMS simulation, Tritscher et al. (2018) show that the observed water vapour is well reproduced.

A potentially missing process that could explain the HCl discrepancy must have the same "fingerprint" as the difference between Figs. 2 and 3 with respect to the timing and dependence on equivalent latitude. In the following section, we focus on the time just after the titration step, when both in the observations and in the model the HCl depletion is still ongoing. Through this we hope to get some pointers about the missing process from looking at these locations and times.

## 5   Characteristics of the HCl discrepancy

To find out more about the characteristics of the potentially missing process that could explain the difference between simulated and observed HCl, we focus on the comparison between simulations and observations for the vortex core data points on the 500 K level for 20 June (data from about 300 MLS profiles). This is a time and location when, in both the model and the observations, the HCl depletion after the initial titration step is ongoing. From that, we can get hints about the possible missing process (or processes).

First, we investigate whether the missing process is correlated with temperature. This would be the case if the depletion of HCl is caused by uptake on PSC particles, for example into the available liquid PSC particles as suggested by the empirical correction in Wohltmann et al. (2017). Similar to $HNO_3$, HCl is soluble in liquid aerosols. The solubility of HCl in the liquid aerosols given by the parameterisation of Carslaw et al. (1995) is included in the CLaMS chemistry module. Although the

5 simulations do not show any significant HCl uptake, it may be that underlying parameters like the Henry's law constant are not accurately represented in the parameterisation as suggested by Wohltmann et al. (2017). The uptake of HCl into the liquid aerosol particles should be strongly temperature dependent (Carslaw et al., 1994), with the largest expected effect for the lowest temperatures. To examine such a hypothesis, the chosen subset of MLS data points and corresponding CLaMS points are plotted as function of temperature in Fig. 7. The temperature for all data points in this subset is below 194 K suggesting

that PSCs should exist at the observation locations shown, which is supported by the MIPAS PSC data base (Spang et al., 2018). The simulations and observations in Fig. 6 suggest that the first titration step is completed by 20 June. The temperature dependence of the difference between observations and the simulation does not, however, suggest a missing uptake of HCl into the particles. The largest discrepancies between simulations and observations are found for higher temperatures. Therefore, the uptake of a significant fraction of HCl into the PSC particles seems to be an unlikely explanation for the discrepancy shown

between Figs. 2 and 3. However, it cannot be excluded that this dependency is caused by a combination of more than one unknown process.

Although the HCl depletion is not directly correlated with temperature, it seems likely that the missing process should require temperatures low enough for heterogeneous chlorine activation on PSCs or cold aerosols. We examine whether the missing process requires sunlight by investigating the history of the same observed air masses using 30-day back-trajectories calculated

by the CLaMS trajectory module. From the trajectories we determined how long the air-mass experienced sunlight and also how long they were exposed to temperatures below 195 K, a typical value below which PSCs can form and heterogeneous chemistry becomes important. Here we used solar zenith angles below 95° for the definition of sunlit time (Rex et al., 2003).

Figure 8 shows the individual observed HCl mixing ratios as a function of the time spent at sunlight and below 195 K as green symbols. Red symbols show the corresponding simulated HCl mixing ratios. There seems to be a clear anti-correlation

between the time spent in sunlight below 195 K and and HCl mixing ratios. This suggests that the missing process may both require sunlight and low temperatures in the presence of PSCs.

To further clarify the temporal development of the HCl discrepancy, Fig. 9 shows also the rate of HCl change dHCl/dt calculated from the time series of HCl within the grid box in the equivalent latitude/potential temperature space for each day (middle panel). It is evident that the first titration step caused by the heterogeneous reaction of HCl with $ClONO_2$ is reproduced

correctly in the simulation. After that step, HCl decreases particularly strongly during two periods in mid-June and mid-July. These periods are correlated with a high occurrence fraction of PSCs as shown in the bottom panel of Fig. 9. Both the HCl depletion rate and PSC occurrence seem to have relative maxima around 11 June (i.e. large rate of HCl depletion), minima around 23 June and again maxima between 5 and 12 July. The PSC occurrence fraction is shown for both MIPAS and CALIOP observations, which are broadly consistent for the overlapping observation times. The corresponding CLaMS PSC occurrence

fractions are also shown. The only significant difference between MIPAS and CALIOP is in late May where MIPAS detects

a large fraction of potentially small STS particles that could not be detected by CALIOP. Also in the beginning, these STS detections are pole-centred and therefore partly missed by the limited latitudinal coverage of the CALIPSO satellite. The anti-correlation between dHCl/dt and the PSC occurrence fraction is a further indication that PSCs may be relevant here. Possible processes missing in the formulation of the model that could explain the HCl discrepancy are investigated in the following
section.

## 6   Potential causes for the HCl discrepancy

The required missing process (or processes) should clearly improve the characteristic difference between observed and simulated HCl mixing ratios (Figs. 2 and 3). By incorporating different processes into a model we can investigate the "fingerprint" of the corresponding change to the model results. This method will not provide a proof of whether a hypothetical process
occurs in reality. However, it can help to exclude hypothetical processes when they cause different "fingerprints" that can be contrasted with observations. In the following subsections, different hypotheses for the missing processes are investigated by incorporation into the model CLaMS.

### 6.1   Ionisation from galactic cosmic rays

One well-known process that is often neglected in simulations of stratospheric chemistry is the ionisation of air molecules
by galactic cosmic rays. Galactic cosmic rays (GCRs) provide an additional source of $NO_x$ and $HO_x$ (Warneck, 1972; Rusch et al., 1981; Solomon et al., 1981; Müller and Crutzen, 1993). The additional source of OH and NO in the polar stratosphere can trigger an HCl sink through the following reaction chains:

$$
\begin{aligned}
NO + O_3 &\rightarrow NO_2 + O_2 \\
NO_2 + ClO &\xrightarrow{M} ClONO_2 \\
HCl + ClONO_2 &\xrightarrow{het} Cl_2 + HNO_3
\end{aligned}
$$

and

$$
\begin{aligned}
OH + O_3 &\rightarrow HO_2 + O_2 \\
HO_2 + ClO &\rightarrow HOCl + O_2 \\
HOCl + HCl &\xrightarrow{het} Cl_2 + H_2O
\end{aligned}
$$

These reaction chains are only effective in the presence of heterogeneously active particle surfaces (cold binary aerosols or PSCs) and in the presence of sunlight since the ClO molecule would be converted into $Cl_2O_2$ in darkness. To investigate this effect, the $NO_x$ and $HO_x$ sources from cosmic-ray induced ionisation pair were incorporated into the chemistry module of CLaMS. The ionisation rate was induced after Heaps (1978) using the efficiency $eff_{NO}$=1.25 and $eff_{HO_x}$=2 for the forma-
tion of NO and OH per ionisation, respectively (Jackman et al., 2016). Fig. 10 shows the corresponding vortex-core average ($\Phi_e$ >75°S) on the 500 K level for the reference simulation and the simulation including induced ionisation by the GCRs. Indeed, including such an ionisation rate in the model does introduce an accelerated HCl decomposition in the polar vortex where

active chlorine exists. However, the effect is too small to explain the CLaMS HCl discrepancy shown in Section 4. By early August, the HCl discrepancy shown here is reduced by about 20%. Even though there are newer estimates of the GCR-induced ionisation rate (Usoskin et al., 2010; Jackman et al., 2016), they are not significantly larger so a severe underestimation of the ionisation rate seems unlikely.

## 6.2 Photolysis of particulate HNO$_3$

Besides the photolysis of gas-phase HNO$_3$, it may be possible that the HNO$_3$ bound in PSC particles also photolyses directly. Evidence for this comes from studies which have shown that nitrate photolysis from snow surfaces is a significant source of NO$_x$ at the Earth's surface (e.g. Honrath et al., 1999; Dominé and Shepson, 2002) and that nitrate photolysis in the quasi-liquid layer in laboratory surface studies is enhanced in the presence of halide ions due to a reduced solvent cage effect (Wingen et al., 2008; Richards et al., 2011). These studies suggest that photolysis of dissolved HNO$_3$ in PSC particles, including STS droplets, may be possible. If such a process could liberate NO$_x$ or HO$_x$ into the gas phase from the particle phase, it could also cause HCl depletion by the reaction chains mentioned above. However, we note that such a process is not yet proven. Wegner (2013) investigated this hypothesis by implementing this process into the model SD-WACCM. He used the cross-section for the NO$_3^-$ ion (Chu and Anastasio, 2003), with a quantum yield of 0.3 reflecting relatively high values seen on some surfaces in laboratory experiments (Abida et al., 2012). These simulations indicated that due to the low solar elevation in austral winters, the onset of HCl depletion in June cannot be reproduced by adding this process. Furthermore, this NO$_x$-source would cause an overestimation of ClONO$_2$ at higher solar elevation in September.

Here we extend on the investigation by Wegner (2013) by including possible upper and lower limits for this process, as the photolysis of particulate HNO$_3$ on PSCs has not been investigated in detail. Laboratory measurements show that combined absorption cross-sections of HNO$_3$(aq) and NO$_3$ in bulk solutions are significantly larger than those in the gas phase (Chu and Anastasio, 2003; Svoboda et al., 2013). Quantum yields in the aqueous bulk phase are small, $\approx$1% or less (Warneck and Wurzinger, 1988; Benedict et al., 2017), increasing to a few percent in the quasi-liquid layers (McFall et al., 2018). However, quantum yields on other surfaces may be much larger (Abida et al., 2012), and such large values are also suggested by strong NO$_x$-emissions from nitrate-containing particles exposed to UV radiation (Reed et al., 2017; Ye et al., 2016; Baergen and Donaldson, 2013). As a lower limit for the photolysis of particulate HNO$_3$ we apply also the gas phase photolysis of HNO$_3$ to the NAT particles with a quantum yield of 1. As an estimate for the upper limit, we use observations of the absorption cross-sections in ice by Chu and Anastasio (2003) and also a quantum yield of 1. The absorption cross-sections employed and the photolysis rates derived from those are shown in Fig. S5 of the Supplementary Material. As in Fig. 10, Figure 11 shows the HCl and ClONO$_2$ data, the CLaMS GCR simulation (blue) and the upper and lower limit as described above. The lower limit (blue lines) is not very different from the CLaMS simulation without this process. In the simulation for the upper limit, an additional faster depletion of HCl is clearly visible. However, throughout the month of June, the effect is not large enough to fully explain the remaining HCl discrepancy. Furthermore, the simulation indicates that this process would significantly enhance the chlorine deactivation into ClONO$_2$ in late August and September. Therefore, the upper limit simulation appears not to be the process that would resolve the HCl discrepancy. Potentially, this process with a lower quantum yield may explain

the small discrepancy in $ClONO_2$ in September. The fingerprint of a photolysis-like process increases with solar elevation and therefore would be much stronger in September than in June. However, there may be a similar processes independent of the solar zenith angle that could decompose condensed-phase $HNO_3$ as discussed in the following subsection.

## 6.3 Decomposition of particulate $HNO_3$

A similar hypothesis to that described above would be the decomposition of the condensed-phase $HNO_3$ by a process other than photolysis. A more steady decomposition throughout the polar winter could be triggered, for example, by GCRs instead of by photolysis. For this hypothesis, sunlight or at least twilight would still be required to form sufficient ClO from the decomposition of the nighttime reservoir $Cl_2O_2$. Over the poles especially, secondary electrons from GCRs that do not have enough energy to decompose air molecules may still be able to interact with $HNO_3$ on the PSCs. In Fig. 12, the HCl discrepancy
in June and July steadily increases with time in the presence of NAT or ice PSCs. To achieve the observed HCl depletion rate on the $500\,K$ potential temperature level, about 1% of the condensed $HNO_3$ per day needs to be liberated into the gas phase. The exact mechanism by which this could occur needs to be clarified, but to investigate the impact of this hypothesis we performed a test simulation in which the $HNO_3$ on NAT is decomposed into $NO_2$+OH at a constant rate of $10^{-7}s^{-1}$, independent of altitude (labelled "NAT decomp").

Figure 12 shows the simulated HCl mixing ratios in $\Phi_e/\theta$-space. Even though not every detail of the observations is reproduced, it is evident that this simulation reproduces the observations much better than any of the hypotheses discussed above. The rate of the hypothetical process was chosen such that the HCl depletion on the $500\,K$ level is well represented. It is evident that above this level this rate should increase. However, it does not seem appropriate to speculate further here.

One further piece of evidence for the hypothesis that a possible $NO_x$-source from PSC decomposition exists is given by the
comparison with $ClONO_2$ data. Figure 12 shows the time development of vortex-core mixing ratios of HCl and $ClONO_2$ on the $500\,K$ potential temperature level. The blue line in the top panel demonstrates that on this level the observed HCl depletion is represented by the simulation "NAT decomp". The MIPAS observations of $ClONO_2$ are intermittent since many observations are blocked by the presence of PSCs. However, the remaining observations indicate low mixing ratios in the range between 0.05 and 0.1 ppbv between late May and late August. These observations are better reproduced by the simulation "NAT decomp"
than both other simulations in which $ClONO_2$ is nearly zero between early June and end of September. From late August to October, simulations of $ClONO_2$ are lower than the simulation. The only sensitivity run that exceeds the observations of $ClONO_2$ is that with the upper limit for photolysis of particulate $HNO_3$ which also demonstrates that an additional $NO_x$-source would increase the $ClONO_2$ mixing ratio.

The change in chlorine activation by this hypothetical process may also be visible in a comparison with MLS ClO, for which
the model output has to be calculated exactly for the location and local time of the observations. For the area of interest defined above (20 June, $\theta$=500 K) the observations in the vortex core are, however, at solar zenith angles such that the ClO mixing ratios are mostly zero within experimental uncertainty. For measurements near the vortex edge, no significant ClO differences are induced by this hypothetical process (compare Fig. S6 and S7 of the Supplementary Material).

The complete fingerprint of the 3-dimensional HCl development similar to Figs. 2 and 3 is shown in Fig. 13. Although differences remain, this sensitivity experiment simulates the HCl observations much better than any other CLaMS simulation discussed here.

## 7   Impact on polar ozone depletion

We now investigate the impact of HCl processing on ozone, since the amount of active chlorine in the polar stratosphere is essential for the determination of chemical ozone loss. All three participating models do simulate the chemical ozone loss and the formation of the ozone hole sufficiently well. Figure 14 shows the simulated vortex average ozone profile at 1 October for the different models in combination with the passive ozone tracer from CLaMS. The difference between the two profiles corresponds to the chemical ozone loss over the winter until this time. The potential process that was included in the simulation "NAT decomp" increases chlorine activation significantly, especially in the polar winter. This might also affect the simulated ozone depletion. However, the difference in chlorine activation is especially large in the rather dark vortex core, where chemical ozone depletion rates are small. The impact of the hypothetical NAT decomposition processes on polar ozone depletion would cause an additional 0.2 ppmv vortex average ozone depletion on the 500 K level as shown in Fig. 14. Figure 15 shows the simulated chemical ozone depletion for the three CLaMS simulations, the reference simulation, the simulation including GCR-induced ionisation and the simulation also including the hypothetical NAT decomposition. The ozone depletion is also shown for the simulation for the Arctic winter 2015/2016 employing the same model setup. Although it is clear that an increase in chlorine activation yields more ozone depletion, the overall effect on ozone depletion of this additional hypothetical process is rather low. The increase of column ozone loss in the vortex core due to GCR-induced ionisation ranges from 2% to 3%. The additional column ozone loss due to the hypothetical NAT decomposition is calculated to be about 1.8% in the Antarctic winter and only 0.6% in the Arctic winter.

## 8   Conclusions

We have demonstrated that an important polar wintertime process which leads to the removal of HCl is missing in the standard formulation of stratospheric chemistry models. The discrepancy between observed and simulated abundances of HCl is apparent in the dark statospheric polar vortex cores in winter. In model simulations there is little change in HCl after the first titration step through the heterogeneous reaction of HCl with $ClONO_2$. In contrast, the satellite observations show a continuous removal of HCl from the gas phase even in the dark core of the polar vortex. We show that this HCl discrepancy is present in three models with different formulations and setups and it is therefore likely not a problem of a single model code. The discrepancy is most clearly demonstrated in the Langrangian CLaMS model, while it is somewhat masked in Eulerian global models likely due to the inherent numerical diffusion of Eulerian transport schemes. A detailed investigation of the characteristics of the HCl discrepancy points to a missing process that likely involves PSCs. Several hypotheses of the cause of the discrepancy have been investigated. The incorporation of GCR-induced ionisation reduces the discrepancy somewhat, but is by no means

sufficient. The uptake of HCl into PSC particles and photolysis of particulate $HNO_3$ also cannot explain the HCl discrepancy. Simulations with a hypothetical steady decomposition of $HNO_3$ out of NAT PSCs would fit the requirements to resolve the HCl discrepancy. This may be caused indirectly by GCRs. At least, the observation of elevated $ClONO_2$ observations in late August supports the presence of an additional $NO_x$-source. However, at present the specific process responsible for the HCl

decomposition remains unclear. Since the discrepancy occurs during the beginning of the chlorine activation period in winter where the ozone loss rates are slow, there is only a minor impact on the overall ozone loss in polar spring.

*Acknowledgements.*    The authors acknowledge the thorough reviews of Ingo Wohltmann and an anonymous reviewer, as well as the comment of Alexander James that helped to clarify and improve this paper. We also thank Susan Solomon and Michael Höpfner for fruitful discussions on aspects of this study. The authors gratefully acknowledge the computing time for the CLaMS simulations granted on the supercomputer

JURECA at Jülich Supercomputing Centre (JSC) under the VSR project ID JICG11. The TOMCAT/SLIMCAT modelling work was supported by the NERC National Centre for Atmospheric Science (NCAS). The simulations were performed on the UK Archer and Leeds ARC HPC facilities. The National Center for Atmospheric Research (NCAR) is sponsored by the U.S. National Science Foundation (NSF). WACCM is a component of NCAR's Community Earth System Model (CESM), which is supported by the NSF and the Office of Science of the U.S. Department of Energy. Computing resources were provided by NCAR's Climate Simulation Laboratory, sponsored by NSF and

other agencies. This research was enabled by the computational and storage resources of NCAR's Computational and Information Systems Laboratory (CISL). The model output and data used in this paper are listed in the references or available from the NCAR Earth System Grid. We also acknowledge the International Space Science Institute (ISSI) for supporting the Polar Stratospheric Cloud Initiative (PSCi). We are grateful to the European Centre for Medium-Range Weather Forecasts (ECMWF) for providing the meteorological re-analyses. Ines Tritscher was funded by the Deutsche Forschungsgemeinschaft (DFG) under project number 310479827. We thank Michelle Santee and the

MLS team, Gabriele Stiller and the MIPAS-Envisat team for the enormous work on providing their high quality data sets.

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

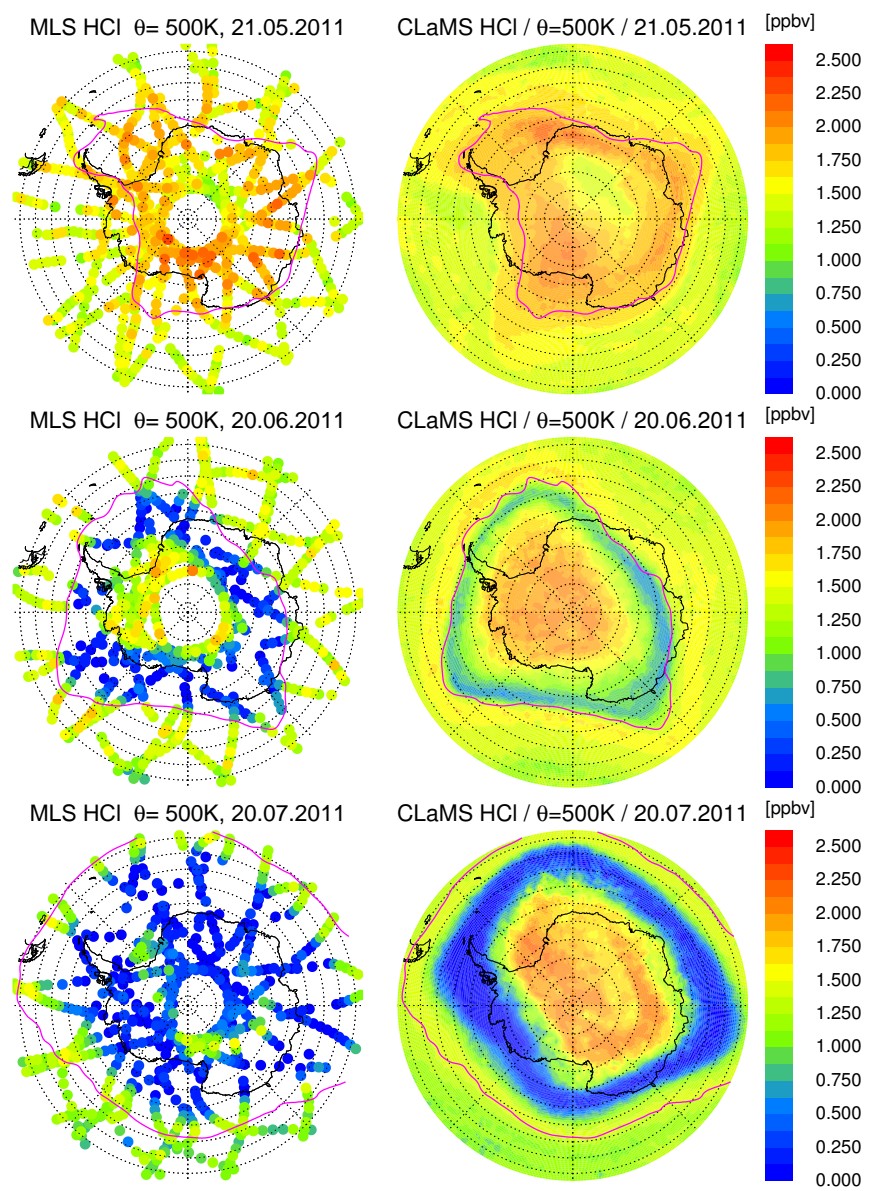

**Figure 1.** Time series of orthographic projection of MLS observations (left) and CLaMS simulations (right) of HCl mixing ratios on the 500 K potential temperature level. The maps are shown for 21 May (top), 20 June (middle), and 20 July 2011 (bottom). The pink line depicts the edge of the polar vortex according to the criterion of Nash et al. (1996).

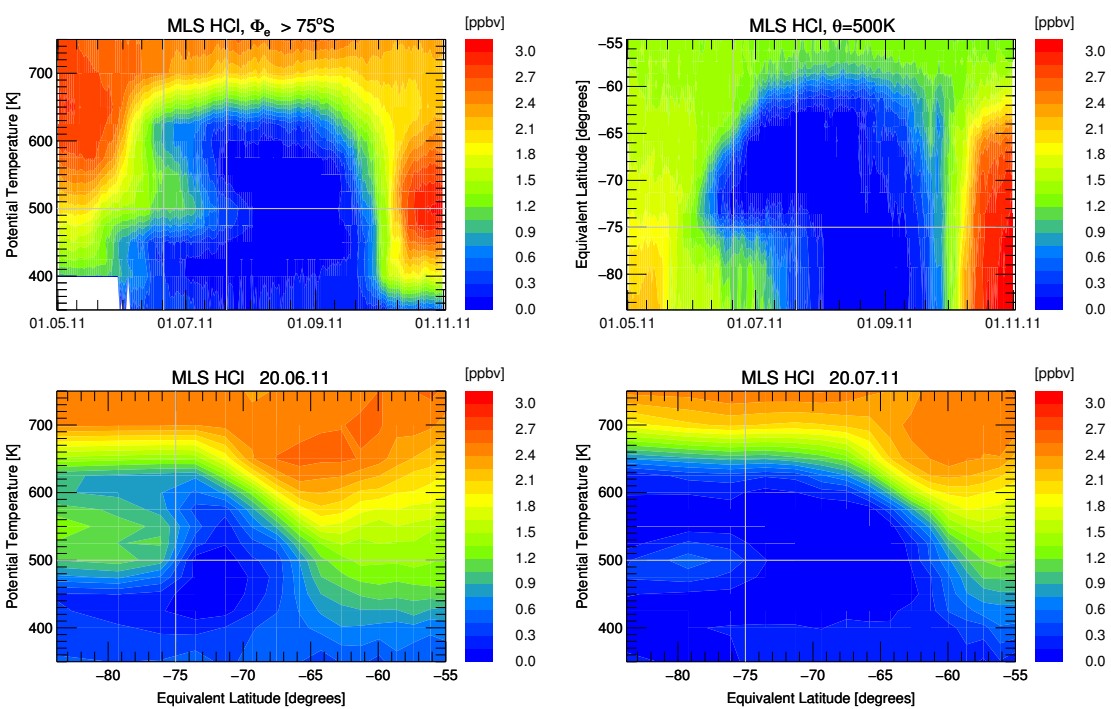

**Figure 2.** Depiction of MLS HCl mixing ratio data averaged in equivalent latitude/potential temperature space. The top left panel shows the vortex-core average for equivalent latitudes poleward of $75°$S as a function of time and potential temperature. The top right panel shows the time development on the 500 K potential temperature level. The bottom two panels show a snapshot of this average on 20 June and 20 July 2011. Grey lines on the panels indicate the cuts or borders displayed in the other panels of this figure.

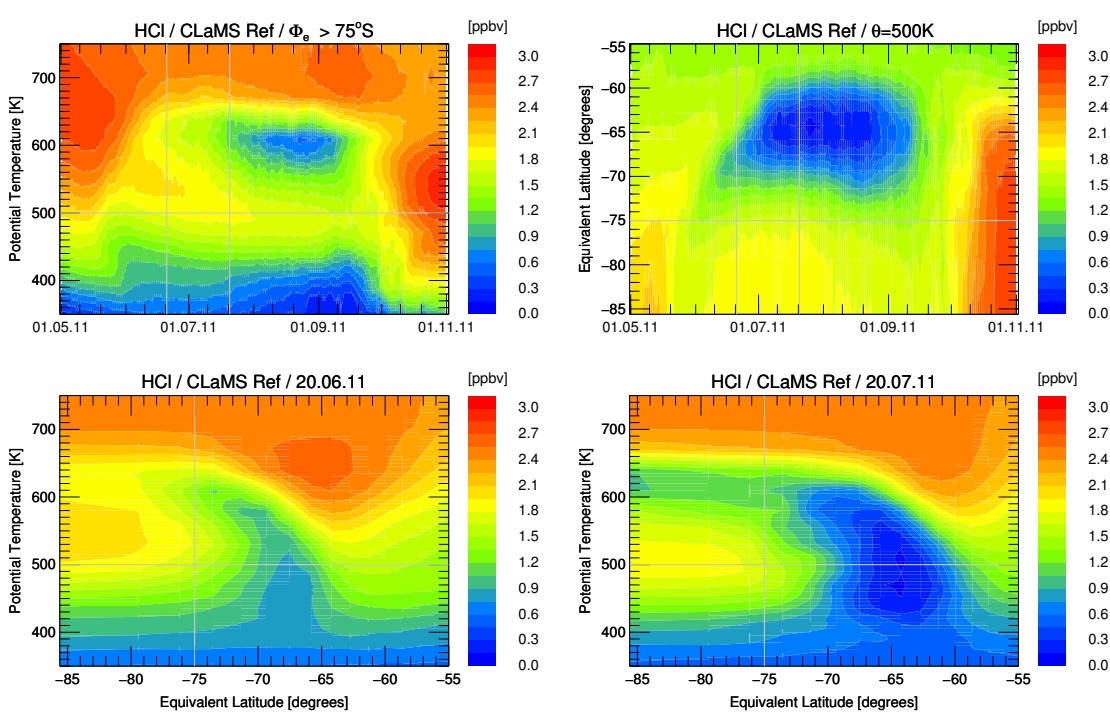

**Figure 3.** CLaMS simulation for HCl averaged in equivalent latitude/potential temperature space displayed as in Fig. 2.

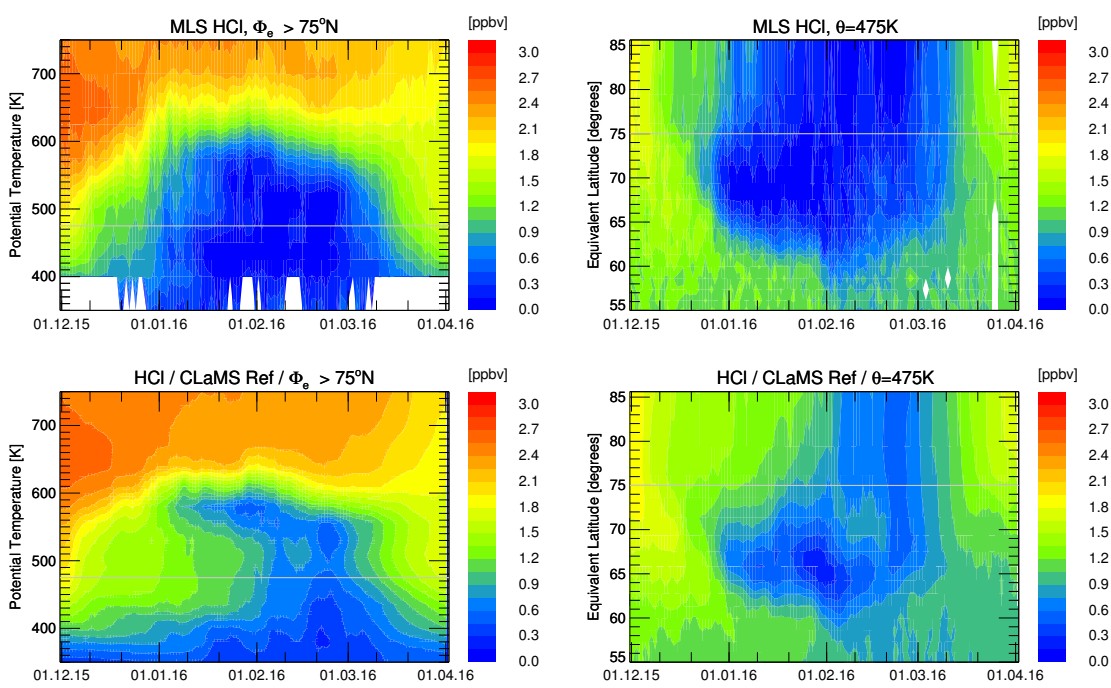

**Figure 4.** Similar results as displayed in Figs. 2 and 3 for the simulation of Arctic winter 2016. The top two panels depict the MLS HCl mixing ratio data averaged in equivalent latitude/potential temperature space for the vortex core ($\Phi_e > 75°$N, top left) and the time development on the 475 K potential temperature level (top right). The bottom two panels show the corresponding model results from CLaMS. Grey lines on the panels indicate the cuts or borders displayed in the other panels of this figure.

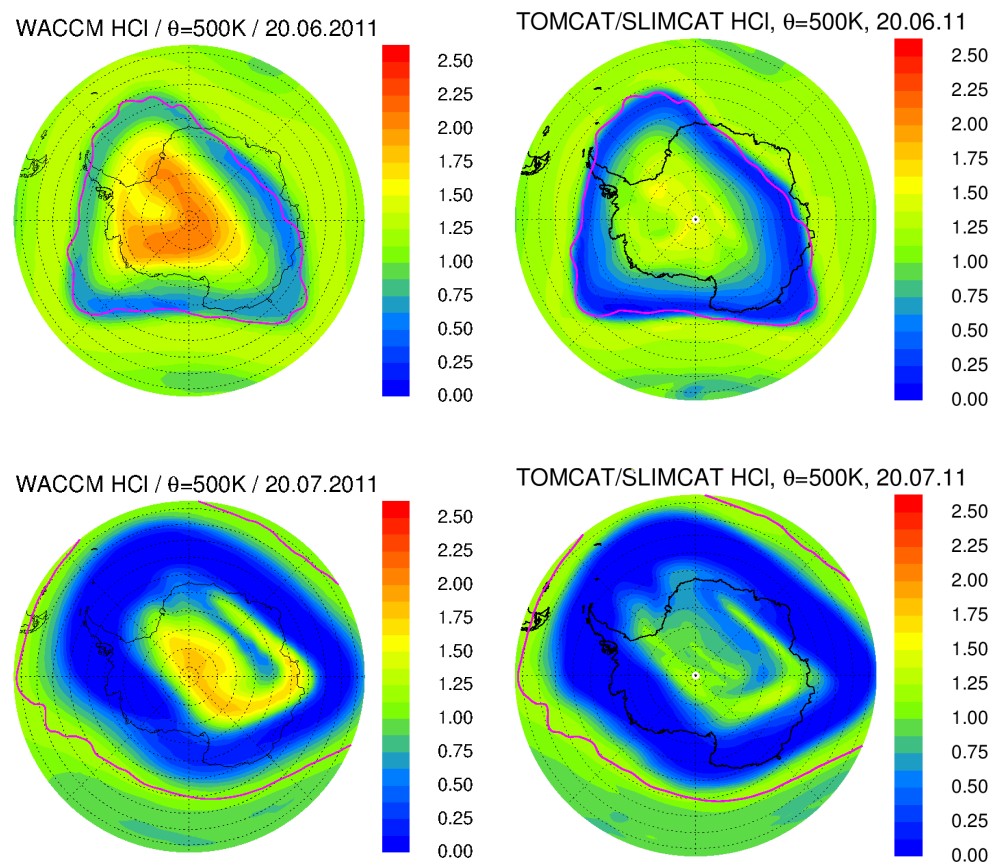

**Figure 5.** Simulated HCl from SD-WACCM (left panels) and TOMCAT/SLIMCAT (right panels) on the 500 K potential temperature surface for 20 June (top panels) and 20 July 2011 (bottom panels) displayed as in Fig. 1.

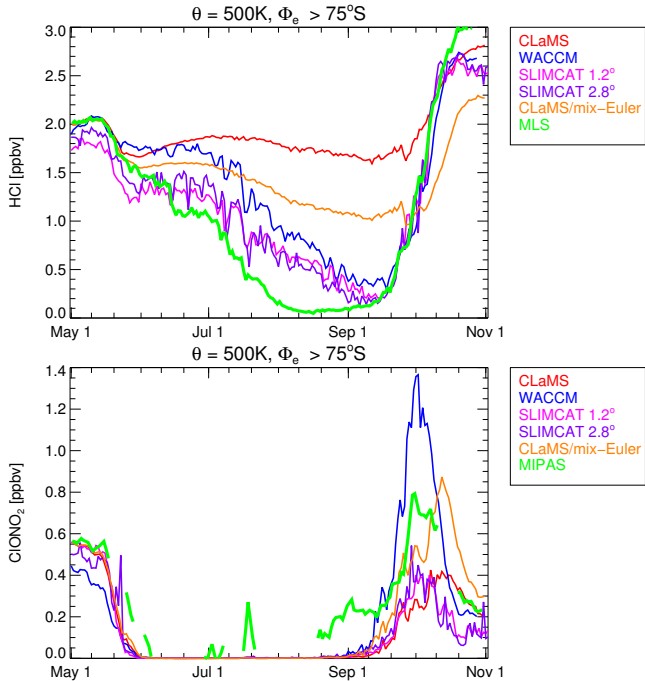

**Figure 6.** Vortex core averages ($\Phi_e > 75°$S) on the 500 K isentrope for HCl (top) and ClONO$_2$ (bottom) for different model simulations. Observations of MLS and MIPAS are shown as green lines. The red line corresponds to CLaMS, the blue line corresponds to SD-WACCM, pink and purple lines correspond to TOMCAT/SLIMCAT simulations with horizontal resolution 1.2° and 2.8°, respectively. The orange line corresponds to the CLaMS simulation mix-Euler, in which additional artificial mixing was applied.

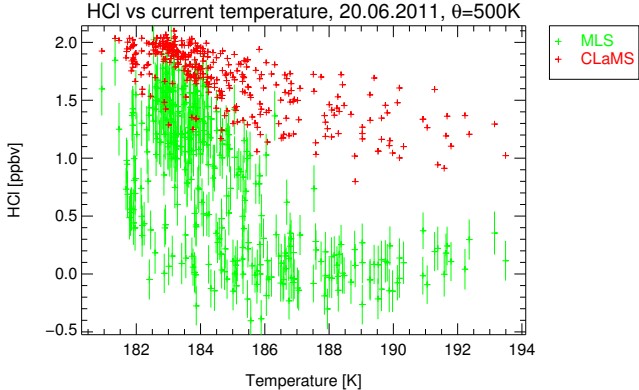

**Figure 7.** MLS observations of HCl mixing ratio on 20 June 2011 in the vortex core ($\Phi_e > 75°$S) on the 500 K potential temperature level. The individual data points are plotted as a function of temperature given by ERA-Interim (green symbols). The error bars show the given measurement precision. The corresponding CLaMS results are given as red symbols.

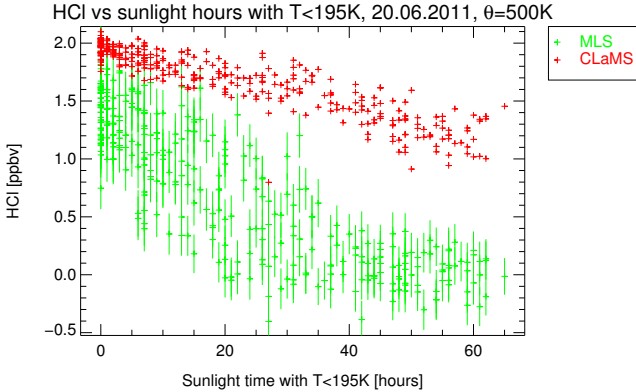

**Figure 8.** MLS observations of HCl mixing ratio that are also displayed in Fig.7 plotted against sunlight time with temperatures below 195 K derived from the backward trajectories (green symbols) and the corresponding CLaMS results (red symbols).

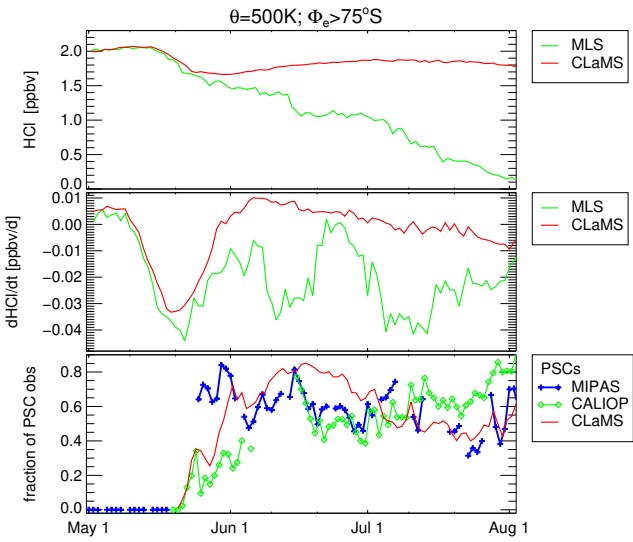

**Figure 9.** Vortex-core averages ($\Phi_e$ >75°S, $\theta$=500 K) of HCl, its time derivative and PSC occurrence frequency. The top panel shows HCl mixing ratios from CLaMS and MLS. The middle panel shows the corresponding time derivative dHCl/dt (smoothed as a 6-day running mean). The bottom panel shows the fraction of MIPAS and CALIOP observations with $\Phi_e$ >75°S on the 500 K potential temperature level that are classified as PSCs. It also shows the correspondng CLaMS results. The symbols are connected with coloured lines except for days without PSC information.

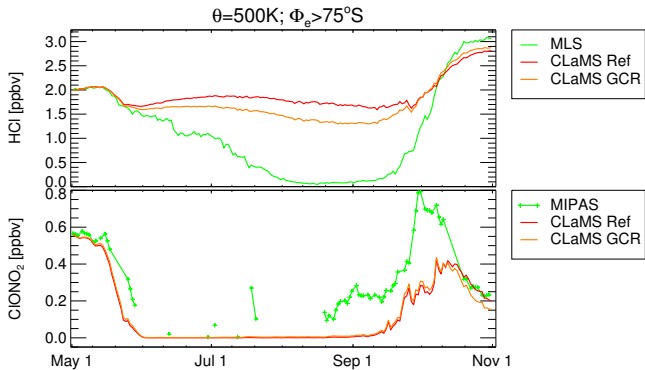

**Figure 10.** Average mixing ratios in the vortex core ($\Phi_e > 75°$ S, $\theta$=500 K) of HCl (top panel) and ClONO$_2$ (bottom panel). Green lines show MLS and MIPAS observations, respectively. CLaMS results are shown for the reference simulation (red) and the simulation with incorporated ionisation from galactic cosmic rays (GCR, orange).

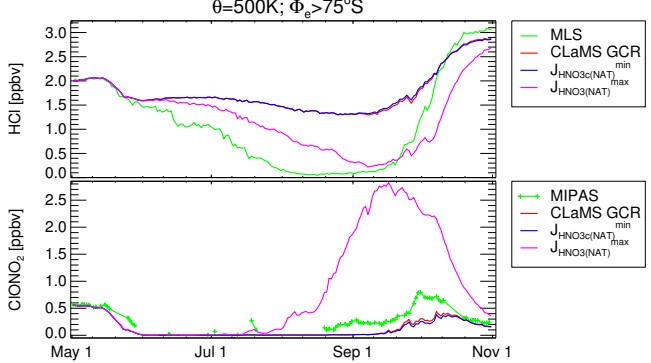

**Figure 11.** Average mixing ratios in the vortex core ($\Phi_e > 75°$ S, $\theta$=500 K) of HCl (top panel) and ClONO$_2$ (bottom panel). Green lines show MLS and MIPAS observations, respectively. The red line shows the CLaMS simulation including GCR-induced ionisation. The blue and pink lines correspond to simulations with additional photolysis of particulate HNO$_3$ using the lower and upper estimate, respectively, as described in the text.

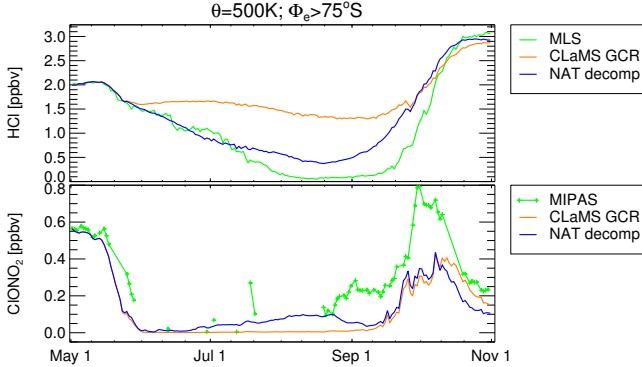

**Figure 12.** As Fig. 10 but for CLaMS simulation with hypothetical NAT decomposition. CLaMS results are shown for the GCR simulation (orange) and the simulation NAT decomp (blue).

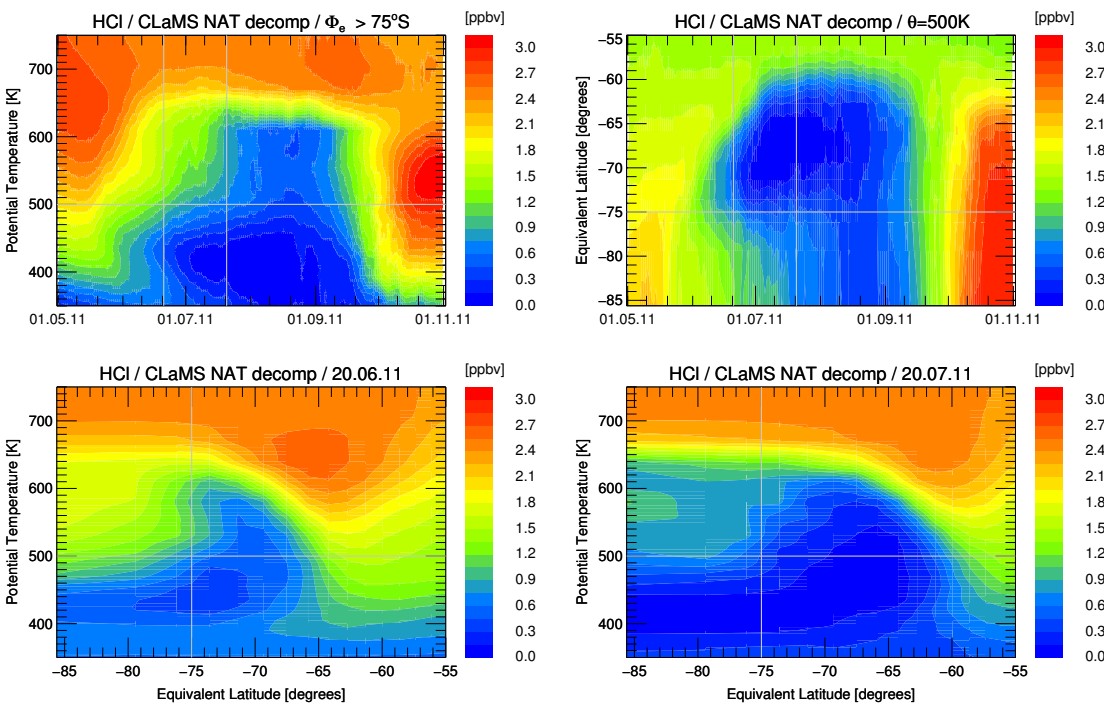

**Figure 13.** CLaMS simulation results for HCl of the sensitivity simulation NAT decomp averaged in equivalent latitude/potential temperature space displayed as in Figs. 2 and 3.

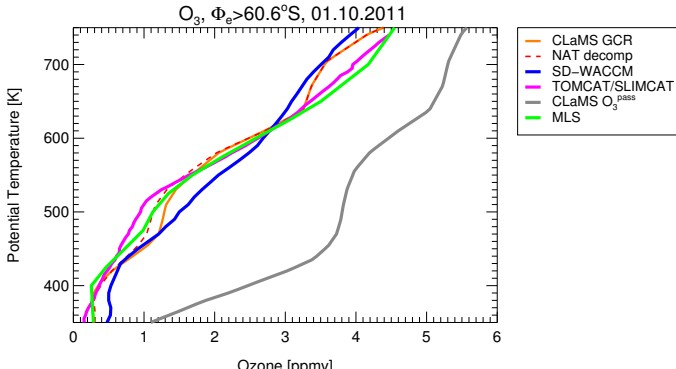

**Figure 14.** Simulated vortex mean ozone profile for 1 October 2011 from MLS (green) and different model runs of SD-WACCM (blue), TOMCAT/SLIMCAT (pink) and CLAMS (GCR, orange). The results from the sensitivity simulation "NAT decomp" are shown as a red dashed line. The grey line corresponds to the CLaMS passive ozone tracer and the difference to the other lines indicates the chemical ozone depletion.

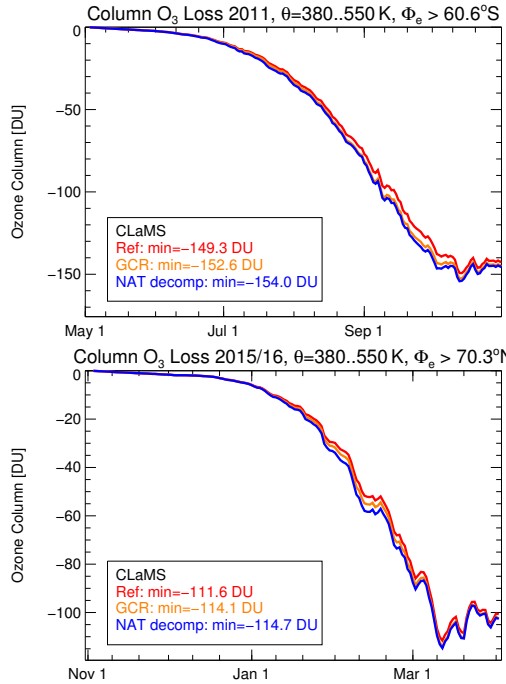

**Figure 15.** Simulated column ozone loss between 380 K and 550 K potential temperature for the Antarctic winter 2011 (top panel) and the Arctic winter 2015/2016 (bottom panel). The results are shown for different sensitivity simulations. Red lines correspond to the reference simulation, orange lines to the simulation including GCR-induced ionisation, and blue lines to the simulation NAT decomp. The legend also indicates the maximum column ozone loss amount of each simulation.