# Peer review of "On the discrepancy of HCl processing in the core of the wintertime polar vortices"

_Atmospheric Chemistry and Physics, 2018_

## Referee Comment (RC1) · I. Wohltmann (Referee) · 13 Apr 2018

Dear Jens-Uwe and co-authors,

this is an interesting paper on an important topic which has been discussed in the stratospheric CTM community for some time now, and I recommend it for publication. Even though this significant HCl discrepancy between models and observations during the onset of chlorine activation does not have a large impact on the overall ozone loss, it may be one of the last remaining gaps in our understanding of stratospheric chemistry and microphysics.

Your explanation based on the decomposition of particulate HNO3 sounds promising to me, and the paper is well structured and written. There are however serious issues with the interpretation of Figure 7 and 8 in section 5 in my opinion (see major comments) and the conclusions drawn from that are probably not valid.

In addition, while I found the paper very detailed and providing much insight on some aspects, in some parts of the paper I miss information that should be in there (literature, other models, other possible explanations), and partly related to this, I don't think that the discussion in the paper is always well balanced. I would recommend to broaden the scope of the paper a little and to summarize what has been discussed so far. See more details in the major comments below.

Ingo Wohltmann

**Major comments**

- I was a little bit surprised that it is not mentioned in the abstract, introduction or conclusion, that there are also other models (apart from the models of the authors) which show the same behaviour (i.e. ATLAS and MIMOSA-CHIM). In addition, some of the relevant literature is not cited. I acknowledge that my Wohltmann et al. (2017) paper is cited later in the detailed discussion, but some other papers are missing completely. I would suggest to cite the following papers in the introduction:

  - Brakebusch et al., J. Geophys. Res., 118, 2673–2688, 2013 (SD-WACCM).
  - Solomon et al., J. Geophys. Res., 120, 7958–7974, 2015 (SD-WACCM). This is only cited in the model description so far.
  - Kuttippurath et al., Atmos. Chem. Phys., 10385–10397, 2015 (MIMOSA-CHIM).
  - Cite Wohltmann et al. (2017) not only in the discussion, but also in the introduction (ATLAS).
  - It may also make sense to cite Santee et al. (2008), J. Geophys. Res., 113, doi:10.1029/2007JD009057 (SLIMCAT), since it shows a discrepancy of opposite sign that may be related to numerical diffusion.

  In fact, the HCl discrepancy seems to be present in all model publications, who have ever looked at HCl mixing ratios in the time period of the onset

of activation in the polar vortex. To me, this strongly suggests that this problem is present in all stratospheric CTMs, even in those who have not published anything on this topic so far. I think it is important to stress this right in the abstract or introduction.

- In a similar direction: Given that this is the first publication which explicitly deals with this topic and given that this topic has been discussed in the stratospheric CTM community for several years, I would expect that this paper gives a better overview over the processes relevant to resolve the discrepancy, which have been discussed so far. These are not restricted to the possible explanations you discuss here in more detail. While you mention some shortly, and some may be easy to discard, I think it would greatly add to the value of the paper, if you would discuss at least some of the following in more detail (following my list in Wohltmann et al., 2017):

    - Initial amount of ClONO2 compared to HCl (how well do we know that?) (e.g. Brakebusch et al., 2013, Wegner, 2013)
    - Over- or underestimation of transport over the vortex edge (e.g. Solomon et al., 2015)
    - Take-up and sedimentation of HCl in cloud particles (Wegner, 2013)
    - Unknown reactions (Wohltmann et al., 2017)
    - An underestimation of the solubility of HCl (the following two bullets are partly are related to this) (e.g. Wegner, 2013, Brakebusch et al., 2013, Wohltmann et al., 2017)
    - A temperature bias in the meteorological data driving the model (e.g. Brakebusch et al., 2013, Solomon, et al., 2015)
    - A water vapour bias in the model

    This is also relevant since the discrepancy will not necessarily have only one reason, but may be caused by a combination of the above. Most of these have already been mentioned in the publications I recommend in the first major comment, and it would be a good idea to summarize the discussion here.

- You discard an underestimation of the HCl solubility as an explanation for the discrepancy all too easily in my opinion (which is also related to issues with the interpretation of your Figure 7, see next major comment). My results show that applying a correction to the solubility greatly helps to bring the MLS and model results in better agreement (see Figures 1 and 2 of this reply, look at the mean running averages). It does not seem to me that your "best explanation" (decomposition of particulate HNO3 with a fixed rate) performs significantly better or that there would be more evidence into this direction. A more balanced discussion would be appropriate here. You certainly do not need to share my opinion, but more discussion of the solubility approach is needed here.

[Figure]

Figure 1: MLS HCl and ATLAS HCl (gas phase) at 46 hPa (one of the MLS measurement levels) as a function of temperature, to compare with Figure 7 of the paper. Top row: Without solubility correction. Bottom row: With solubility correction as applied in Wohltmann et al. (2017). Columns are different dates (21.5.2011, 20.6.2011, 20.7.2011). Solid lines are moving averages (MLS black, ATLAS dark green). All measurements south of 30 deg S are shown.

[Figure]

Figure 2: Same as above, but for 31 hPa.

[Figure]

Figure 3: Same as in Figs. 1 and 2 for 15.7.2006 at 46 hPa, color coded by water vapor mixing ratio (crosses for modelled HCl and H2O for ATLAS, dots measured for MLS).

I am the first to acknowledge that the solubility approach is certainly not "the" explanation or the only cause of the discrepancy and that it has several serious issues. For example, while the running averages (as a function of temperature) agree surprisingly well after application of my 5 K correction (Figs. 1 and 2), individual points show large deviations.

To illustrate a particularly bad example, I show data from 15 July 2006 in Figure 3. It is evident that under dehydrified conditions under very cold temperatures, the solubility approach does a very bad job. This is related to the fact that the solubility decreases with decreasing water vapor. However, in the MLS data, very low HCl levels are maintained under dehydrified conditions, excluding the solubility approach as an explanation under dehydrified conditions with the current parameterization.

In addition, the 5 K shift is really huge and not compatible with the Luo et al. (1995) parameterization that we probably all use in our models. The solubility parameterization is however not really my area of expertise, and it is difficult to judge for me in how far a different parameterization would be possible and be of help here. Maybe it would be an idea to both introduce a shift in temperature and in water vapor?

I think it would be important to discuss the things I mention in the above paragraphs in a little more detail to get a more balanced discussion.

- I think there are serious issues in the interpretation of your Figure 7 and 8 in section 5. I don't think you can draw the conclusions about the temperature dependency and dependency on sunlight of the HCl discrepancy that you draw here and that this section has to be rewritten.

Small changes to the altitude or the date of the figure produce results that can be interpreted completely differently. As an example, I have shown results to compare with your Figure 7 for 46 hPa (my Figure 1) and 31 hPa (my Figure 2) (the 46 hPa and 31 hPa levels are measurement levels of MLS roughly enclosing your 500 K level). These show a completely different temperature dependency for the MLS data for 20.6.2011, which is related to the "tongue" of high HCl visible in Figure 2 of your paper. While the 46 hPa data show a decrease of MLS HCl with decreasing temperature (upper row, second column, Figure 1), the 31 hPa data show an increase. Similar differences in interpretation are found when looking at another date (20.7.2011, right column).

Apart from this vulnerability to small changes, I think that this kind of figure is notoriously difficult to interpret, if a) there is more than one process causing the discrepancy, and b) the processes changing HCl have a memory. I.e., the figure is only easy to interpret if HCl is determined instantaneously by a single process.

The same issues apply to your Figure 8. I can easily produce a figure, where with increasing sunlight time, the discrepancy does not increase (Figure 4 of this review).

But that is even not the main problem with your Figure 8. When I try to reproduce your Figure 8 with the 195 K constraint, I get much lower sunlight hours than you, and no apparent dependency of the discrepancy on sunlight hours below 195 K (my Figure 4, left). When I however just calculate the total sunlight hours (right), the figure looks more similar to your figure (note that I plotted all measurements south of 60 deg S, which explains the higher values for the sunlight hours). I have the impression that there may be a bug in the calculation of the values for your Figure 8, and that the 195 K constraint may be missing.

That may also explain why your Figure 7 and 8 look so similar, since temperature and sunlight hours are highly correlated if you skip the 195 K constraint (Figure 5 of this review).

- You spend quite some time to show that the process is dependent on sunlight in section 5. But your best explanation (6.3, decomposition of particulate HNO3) is not dependent on sunlight. How does this fit together? This obviously requires some discussion in the paper. Maybe it is related to the issues in section 5.

**Specific comments**

- Page 1, line 9: Wouldn't it be better to speak of SD-WACCM (here and elsewhere) to make clear that you are using the specified dynamics version of WACCM here?

[Figure]

Figure 4: HCl as a function of sunlight hours below 195 K (left) and total sunlight hours (right), for 30 day back trajectories starting 20.6.2011 at the MLS measurement locations at 46 hPa south of 60 deg S for my uncorrected run, to compare with your Figure 8. MLS is green and ATLAS is red.

[Figure]

Figure 5: Sunlight hours below 195 K (left) and total sunlight hours (right) as a function of temperature at start for 30 day back trajectories starting 20.6.2011 at the MLS measurement locations at 46 hPa south of 60 deg S.

- Page 1, lines 16–17: Really? You discard this all too easily in my opinion, see major comment.

- Page 1, lines 20: As long as you don't know the processes that would lead to this HNO3 decomposition, this is no better explanation than the solubility approach.

- Page 2, line 30–31: "not yet been reported". You need to change the formulation. In the next sentence, you show that this is not true and that it has been reported in Wegner et al. (and later in Brakebusch et al., Solomon et al., etc.). Maybe just keep the next sentence and delete this one?

- Sections 2.1–2.3: I would suggest to describe the different heterogenous schemes (microphysics/chemistry/sedimentation) in some more detail here (as long as it is important for HCl) and to contrast the schemes of the different models to point out in which respect they are different (or identical). That may also be helpful to understand differences in HCl between the models better.

- Page 7, lines 13–15: I find the tone of this discussion unnecessarily negative. I am certainly not a completely neutral reviewer with respect to my model, but I think a neutral reviewer would have the same comment. First, you write that I did not "find an explanation". But the same is true for this paper. Both of us examined several possibilities for explanations of the discrepancy, and both of us did not find a "final" explanation that would resolve the issue completely. Then, your "best" explanation is as "empirical" as mine. The text here and the following discussion in the next sections however sounds as if my explanation would be empirical and arbitrary, while this is not the case for your "best" explanation. Finally, the addition "but without further evidence why that could be." is completely unnecessary. Please delete this part. I discussed the pros and cons of different explanations on more than two pages, even though I was restricted to keep this short for several reasons. You have very similar problems in your paper ("...the exact mechanism needs to be clarified ...")

  I would suggest a more balanced discussion here, at least that it is mentioned that you have similar problems to the ones that I faced in my publication. A more balanced discussion would also include to mention ATLAS (and the Kuttippurath paper for MIMOSA-CHIM) in the introduction and/or conclusions.

- Page 8, line 5: You write "For this example trajectory". Does that mean that there are other cases which look worse? Or is the trajectory representative?

- Page 9, line 8: It is also worth mentioning that it is also strongly dependent on water vapor.

- Page 9, lines 12–15: See major comment. I don't think you can draw this conclusion.

- Figure 7: It is interesting that both the CLaMS and the "uncorrected" ATLAS HCl values seem to increase with decreasing temperature. As I said in the major comment, I would not overinterpret a plot like Figure 7, but maybe that hints to something?

- Page 9, line 16–24, Figure 8: See major comment. I don't think you can draw this conclusion.

- Page 9, line 29 and Figure 9: I can't follow you here that there is a correlation between dHCl/dt and PSC occurence based on Figure 9. While there are clear minima in dHCl/dt, the PSC occurence oscillates somewhere around 0.6 all the time. In fact, there are e.g. higher values for PSC occurence in end of May than during the periods of low dHCl/dt values later in June and July. It is also not clear to me what the discussion in lines 25–35 is supposed to tell me. I have the impression that this needs to be rephrased.

- Section 6.3: This looks quite promising and very interesting. I would however make more clear somewhere in the paper that there may be other explanations, which possibly work equally well, and which are not discussed in this manuscript. In particular, while your argumentation leading to this explanation is quite logical and sound, the weak point is your assumption of a rather arbitrary fixed rate here, where you don't really know where it comes from. That is quite comparable to the situation with my ad hoc assumption of a 5 K shift in HCl solubility, where I don't really know where it comes from.

- Section 6.3: How does the simulated HNO3 compare to MLS HNO3, with the decomposition switched on and off? A better agreement when switching on the decomposition would strengthen your point.

- Page 12, line 10: Is the rate of $10^{-7}$s$^{-1}$ compatible with the amount of galactic cosmic rays (or secondary electrons with the right energy)? Or is this just a value empirically chosen to get the best fit with the HCl measurements? That should be stated here.

**Technical corrections**

- Page 9, line 22: "in sunlight"?

- Page 11, line 4: Change "Evidenc" to "Evidence"

- Page 11, lines 32–33: Change "a similar processes" to "a similar process"

---

## Referee Comment (RC2) · Anonymous Referee #2 · 20 Apr 2018

**On the discrepancy of HCl processing in the dark polar vortices**

*By Grooss et al.*

**General:**

The difference between measured and simulated HCl in the polar stratosphere has been a problem for modelers and also for chlorine portioning studies. The authors address this issue in this manuscript and it merits a discussion. However, more background information on previous modeling works on this subject and HCl/ClO comparisons should be presented in detail. The manuscript is well written at parts and hence, a language editing is also needed. Please find my specific comments below. The manuscript can be accepted for a publication after this minor revision.

**Specific comments:**

1. Introduction has to be elaborated with previous modeling studies in the polar stratosphere and HCl comparisons (e.g. Feng et al., Wholtmann et al., and Kuttippurath et al. articles on polar processing and ozone loss studies)

2. As stated in the introduction, the main idea was to check the impact of HCl discrepancy on ozone loss or polar ozone loss chemistry. However, that section is too short and limited to the description of the impact of change in ozone with respect to different model experiments. I would suggest you to calculate the ozone loss (profiles too) and compare with the published results (even for similar winters in the past). This would also give an idea about the model performance in comparison to other models.

3. I think that you missed ClO comparisons in this study, although you have a comparison with $ClNO_2$. You have described a lot about the chlorine partitioning and chemical polar processing (e.g. page 12, line 28—29). Therefore, I think it is important to compare the simulated ClO (from different experiments) with measurements (e.g. from MLS).

4. Page 8, Para 2: You stated that the numerical diffusion masks the HCl differences in Eulerian models. However, still the HCl discrepancy is very much apparent in those models/simulations, as demonstrated in this manuscript? So how much is the contribution from numerical diffusion?

5. You have used three different models for this study, which is also the strength of this study. However, a discussion on the ability of synergetic use of the models to be applied for such studies is missing here. Only different test simulations are given. Please include a brief discussion in **Section 6**, and add few lines in conclusions too.

**Technical:**

Line 8: and, to date, to varying
Line 22: rates are small

Line 12: a major role

Line 6: data from the
Line 8: the model results and comparison
Line 9: mixing process related to HCl, and
Line 9: " sh o w"

Line 2: resolution were
Line 16: we use the MLS
Line 17: You did not use ClO data?
Line 24: Use " However ," i n s te a d o f " unfortunately" t un a
Line 26: delete "unfortunately"

Line 11: comparison to other models is absent in this section (e.g. MIMOSA-CHIM, REPROBUS, ATLAS, etc.)
Line 23: latitude calculated from the ERA-interim
Line 27: " However, the ClONO$_2$ observations"
Line 31: " underestimate"
Line 32: all simulations? From all models?

Page 8,
Line 7: " dark unmixed polar"
Line 7: differences
Line 8: " not likely", use "are unlikely....."
Line 8: model differences
Line 32: Is there any reasons for taking 500 K altitude for this comparison?

Line 6: show any significant
Line 18: Delete " Therefore"
Line 30: CALIOP observations, which

Line 4: Evidence
Line 14: overestimation
Line 23 and 24: "cross-sections" would be better in this context

Line 26: This must be section 7
Line 33: the same model setup

Page 13:
Line 9: Numerical diffusion! Then how can we use these models even for this study (e.g. HCl differences)?
Line 18: How much is this minor? Significant?

---

## Short Comment (SC1) · 27 Apr 2018

Comment on "On the discrepancy of HCI processing in the dark polar vortices"

Alexander D. James

School of Chemistry, University of Leeds, Leeds LS2 9JT, UK

The authors present a useful summary, and thought-provoking new ideas, relating to a relatively well-known problem in stratospheric chemistry. This is likely to provoke future research and certainly merits publication following some revision.

The reference to the "dark polar vortex" in the title seems misleading given that the authors show an apparent correlation of the HCI discrepancy to hours of sunlight.

The reviews presented in RC1 and RC2 detail a number of additional processes which should be discussed in the text. In particular the paper lacks a discussion of the possible role of solid PSC. The loss of HCl is known to proceed rather differently on nitric acid hydrate (NAX) or ice surfaces compared to liquid PSCs (Chu et al., 1993). Hoyle et al. (2013) recently showed a good agreement between CLAMS and CALIPSO data on PSC type when a heterogeneous nucleation scheme was implemented, leading to a sharp onset in NAX nucleation below 195 K. In James et al. (2018) we then showed, based a study in our laboratory, that such heterogeneous nucleation can be caused by meteoric smoke or fragmented meteoroids, and can occur at temperatures as high as 197 K for typical stratospheric abundances of HNO3, H2SO4 and H2O.

In figure 7 of the current text, the discrepancy is shown not to relate to temperatures below 194 K. I would like to make several points about this approach. Firstly, there is not only a lack of correlation between the model and observations, but in fact the general trend of HCI concentrations is rather different. The model shows a gradual increase in HCI at lower temperatures, whereas the observations show a significant increase below 186 K. This temperature is consistent with the formation of water ice PSC. It would also be informative for the reader if data were included to higher temperatures, to include the formation of possible NAX between 197 and 194 K.

In producing Figure 8 (setting to one side the concerns raised by the reviewers), a cut-off temperature of 195 K is used. This is reasonable if the growth of liquid PSC through deposition of H2O is the determining factor; however, it is known that significant nucleation of NAX can occur above 195 K (Peter and Grooß, 2012). A more appropriate approach might be to examine the difference between the nucleation scheme used in each model (presumably the same as the Hoyle et al. (2013), Carslaw et al. (2002) and Brakebusch et al. (2013) schemes for CLAMs, SLIMCAT and WACCM respectively) and the more recent literature. In James et al. (2018) we showed that different parameterisations of nucleation give rather different results in terms of particle concentrations and therefore surface area.

A simple and informative experiment would be to correlate the HCI discrepancy to the difference between the onset of nucleation (e.g. time spent between 197 and 195 K for models using a constant volume nucleation rate, see Figure 6 in James et

al. (2018)). Since MIPAS data are used extensively another obvious approach would be to relate the HCI (or CIONO2) discrepancy to the data on PSC type which MIPAS provides.

It is not possible given the present text to assess whether the presence of solid PSC could explain the discrepancies; however, with the inclusion of the references mentioned above and the subsequent suggestions (see above), this could prove an important process contributing to the HCI discrepancy.

Brakebusch, M., Randall, C. E., Kinnison, D. E., Tilmes, S., Santee, M. L., and Manney, G. L.: Evaluation of Whole Atmosphere Community Climate Model simulations of ozone during Arctic winter 2004–2005, J. Geophy. Res.: Atmos., 118, 2673-2688, 10.1002/jgrd.50226, 2013.

Carslaw, K. S., Kettleborough, J. A., Northway, M. J., Davies, S., Gao, R.-S., Fahey, D. W., G., B. D., Chipperfield, M. P., and Kleinböhl, A.: A vortex-scale simulation of the growth and sedimentation of large nitric acid hydrate particles, J. Geophys. Res.: Atmos., 107, SOL 43-41-SOL 43-16, doi:10.1029/2001JD000467, 2002.

Chu, L. T., Leu, M. T., and Keyser, L. F.: Uptake of hydrogen chloride in water ice and nitric acid ice films, J. Phys. Chem., 97, 7779-7785, 1993.

Hoyle, C. R., Engel, I., Luo, B. P., Pitts, M. C., Poole, L. R., Grooß, J. U., and Peter, T.: Heterogeneous formation of polar stratospheric clouds – Part 1: Nucleation of Nitric Acid Trihydrate (NAT), Atmos. Chem. Phys., 13, 9577-9595, 2013.

James, A. D., Brooke, J. S. A., Mangan, T. P., Whale, T. F., Plane, J. M. C., and Murray, B. J.: Nucleation of nitric acid hydrates in polar stratospheric clouds by meteoric material, Atmos. Chem. Phys., 18, 4519-4531, 10.5194/acp-18-4519-2018, 2018.

Peter, T., and Grooß, J. U.: Chapter 4: Polar stratospheric clouds and sulfate aerosol particles: Microphysics, denitrification and heterogeneous chemistry, in: Stratospheric Ozone Depletion and Climate Change, The Royal Society of Chemistry, 108-144, 2012.

---

## Author Comment (AC1) · 22 May 2018

We thank Ingo Wohltmann for his constructive review that helped us to improve the paper. Some of the raised points may be solved by a better explanation in the revised manuscript, but others pointed us to necessary modifications and additions to our paper. The comments of Ingo Wohltmann are numbered and repeated below indented and *in italic letters* followed by our answers.

Major comments

> **1.** *I was a little bit surprised that it is not mentioned in the abstract, introduction or*
> *conclusion, that there are also other models (apart from the models of the authors)*

[Figure]

*which show the same behaviour (i.e. ATLAS and MIMOSA-CHIM). In addition, some of the relevant literature is not cited. I acknowledge that my Wohltmann et al. (2017) paper is cited later in the detailed discussion, but some other papers are missing completely. I would suggest to cite the following papers in the introduction:*

- *Brakebusch et al., J. Geophys. Res., 118, 2673-2688, 2013 (SD- WACCM).*
- *Solomon et al., J. Geophys. Res., 120, 7958-7974, 2015 (SD-WACCM). This is only cited in the model description so far.*
- *Kuttippurath et al., Atmos. Chem. Phys., 10385-10397, 2015 (MIMOSA-CHIM).*
- *Cite Wohltmann et al. (2017) not only in the discussion, but also in the introduction (ATLAS).*
- *It may also make sense to cite Santee et al. (2008), J. Geophys. Res., 113, doi:10.1029/2007JD009057 (SLIMCAT), since it shows a discrepancy of opposite sign that may be related to numerical diffusion.*

Thank you for these suggestions. Although we did realise that other models seem to have the same problem, we admit that were not aware that these are present and partly discussed in publications. Besides in the Wohltmann paper, the discrepancy is mostly not clearly addressed. In the revised version, we follow these suggestions and add the following paragraph to our introduction section:
"This discrepancy, hereafter referred to as the "HCl discrepancy", has been shown in previous publications, although it was mostly not the focus of those studies. Brakebusch et al. (2013) show the HCl discrepancy in a simulation for the Arctic winter 2004/2005. It could be partly corrected in the vortex average by decreasing the temperature in the module for heterogeneous chemistry by 1 K. Solomon et al. (2015) also show the discrepancy in SD-WACCM for early winter 2011. It is present at 82°S and

53 hPa (their Fig. 4) and at 80°S and 30 hPa (their Fig. 8) in all of the sensitivity studies shown. However, the focus of that paper was on the late winter and spring period and the issue was not discussed further. Kuttippurath et al. (2015) show 10 years of simulation with the model MIMOSA-CHIM. They compare the time dependence of vortex average mixing ratios with MLS observations and present an average of the 10 Antarctic winters 2004-2013. In their Fig. 4, the discrepancy also seems to be present, even though it is smoothed out by the averaging procedure. Recently, Wohltmann et al. (2017) did explicitly address the HCl discrepancy. They show simulations with the Lagrangian model ATLAS in comparison with MLS observations for the winter Arctic 2004/2005. Although the comparison with the other chemical compounds is very good, for example with MLS $N_2O$ and ozone, the observed HCl mixing ratios also indicate a depletion in early winter that is not present in the vortex-average simulation shown. As a possible solution they suggest an increased uptake into the liquid STS particles due to a higher solubility by imposing an artificial negative temperature offset of 5 K. With that, the vortex average HCl mixing ratios decreased more in early winter. However, there is no evidence for what the reason of this enhanced solubility could be. Santee et al. (2008) also show a MLS-model comparison of HCl with apparently the opposite problem, that is a modelled depletion of HCl before it was observed and with a larger vertical extent. These simulations were performed by an earlier version of the TOMCAT/SLIMCAT model and a simple PSC scheme that, for example, triggers PSC formation directly at the NAT equilibrium temperature. Further, the initial $ClONO_2$ in this study could not be constrained with observations. We therefore concentrate here on the simulations with the updated version of TOMCAT/SLIMCAT."

> **2.** *In fact, the HCl discrepancy seems to be present in all model publications, who have ever looked at HCl mixing ratios in the time period of the onset of activation in the polar vortex. To me, this strongly suggests that this problem is present in all stratospheric CTMs, even in those who have not published anything on this topic so far. I think it is important to stress this right in the abstract or introduction.*

We suggest in our manuscript that there should be an unknown process that is missing in the formulation of current models and it may seem likely that this discrepancy *"is present in all stratospheric CTMs".* As stated above, we added more citations to studies, in which the HCl discrepancy was mentioned or is at least present in figures. However, we cannot prove this statement and therefore would refrain from stressing this point too much. But the added citations and statements (see point 1.) will show that the HCl discrepancy is present in other models as well.

**3.** *In a similar direction: Given that this is the first publication which explicitly deals with this topic and given that this topic has been discussed in the stratospheric CTM community for several years, I would expect that this paper gives a better overview over the processes relevant to resolve the discrepancy, which have been discussed so far. These are not restricted to the possible explanations you discuss here in more detail. While you mention some shortly, and some may be easy to discard, I think it would greatly add to the value of the paper, if you would discuss at least some of the following in more detail (following my list in Wohltmann et al., 2017):*

- *Initial amount of $ClONO_2$ compared to HCl (how well do we know that?) (e.g. Brakebusch et al., 2013, Wegner, 2013)*
- *Over- or underestimation of transport over the vortex edge (e.g. Solomon et al., 2015)*
- *Take-up and sedimentation of HCl in cloud particles (Wegner, 2013)*
- *Unknown reactions (Wohltmann et al., 2017)*
- *An underestimation of the solubility of HCl (the following two bullets are partly are related to this) (e.g. Wegner, 2013, Brakebusch et al., 2013, Wohltmann et al., 2017)*
- *A temperature bias in the meteorological data driving the model (e.g. Brakebusch et al., 2013, Solomon, et al., 2015)*

- *A water vapour bias in the model*

*This is also relevant since the discrepancy will not necessarily have only one reason, but may be caused by a combination of the above. Most of these have already been mentioned in the publications I recommend in the first major comment, and it would be a good idea to summarize the discussion here.*

Thank you also for these suggestions. We included more discussion about these issues to the revised version of the paper.

(a) The initial $ClONO_2$/HCl ratio is indeed important. If these compounds are not initialised correctly, this will have immediate impacts on the first titration step. In Fig. 6 of the paper the titration step is visible in the differences between the different model formulations. The advantage of CLaMS in this respect is that the model is initialised by MLS HCl and MIPAS $ClONO_2$ data.

(b) transport though the vortex edge and also mixing within the vortex is an issue. We argue, based on our simulations that artificial numerical diffusion may be the cause for an increase in HCl depletion rates.

(c) The possible uptake of HCl into PSC particles has been discussed. A significant HCl loss through sedimentation seems unlikely, since a permanent HCl removal would inhibit the HCl increase at the end of the winter and would thus lead to an inconsistency that is not observed. This will be mentioned in the revised version.

(d) Unknown chemical reactions: Wohltmann et al. (2017) briefly discuss an additional heterogeneous reaction involving HCl and conclude that this cannot be excluded but it must fulfil the conditions that it does not change much the remaining chemical composition. We agree with this statement and will repeat it here.

(e) An under-estimation of the solubility of HCl would lead to an increased uptake of HCl into the particles, see point (c)

(f) A temperature bias of the underlying meteorological analyses would in principle be possible. However, during the period of initial chlorine activation it would likely not have

a significant impact, as the HCl depletion is limited by the availability of ClONO$_2$, not by the strongly temperature dependent heterogeneous reaction rate.

(g) Water vapour bias: Tritscher et al. (ACPD, 2018) show a comparison of both H$_2$O and HNO$_3$ with MLS observations. There seems to be no significant H$_2$O bias in the CLaMS simulation (see Fig. 3 of this reply).

> **4.** *You discard an underestimation of the HCl solubility as an explanation for the discrepancy all too easily in my opinion (which is also related to issues with the interpretation of your Figure 7, see next major comment). My results show that applying a correction to the solubility greatly helps to bring the MLS and model results in better agreement (see Figures 1 and 2 of this reply, look at the mean running averages). It does not seem to me that your "best explanation" (decomposition of particulate HNO$_3$ with a fixed rate) performs significantly better or that there would be more evidence into this direction. A more balanced discussion would be appropriate here. You certainly do not need to share my opinion, but more discussion of the solubility approach is needed here.*

First, we agree that the discussion of the solubility issue should be more extensive and more balanced in the revised version of the manuscript and as expressed above, we did do so. To best find out the possible mechanism, we did subset the MLS data to a time altitude and equivalent latitude, where both in MLS and the model the HCl depletion is still ongoing. With that, we expect to find hints to the possible missing process in correlating this with relevant parameters. This would work less efficiently when the HCl is already completely depleted. At different latitudes, altitudes or times, the discrepancy between model and observations likely shows other temperature dependencies, but we believe that one may learn less from that for the possible missing process.

Thank you for the work of also plotting similar plots with the model ATLAS. We do think that it is valid to show Figure 7, but we should add some clearer arguments in the

revised version. We will also discuss the HCl solubility to some more extent.

In principle, you plotted a super-set of the data in our paper. In the cold Antarctic vortex in June, 31 hPa roughly corresponds to 500 K, whereas 46 hPa roughly corresponds to 450 K. The Figs. 1 and 2 of this comment are the corresponding plots to 46 hPa ($\theta$=450 K) that also indicate the absence of a significant dependence at this altitude.

In the so defined time and space range (June, 500 K/31 hPa, $\Phi_e > 75°$S), it is evident both in CLaMS and ATLAS that for very low temperatures ($\approx$180-185 K) the discrepancy is smaller than for higher temperatures ($\approx$185-190 K). A larger HCl solubility has a larger effect at lower temperatures just as shown in the corresponding ATLAS plot, where the additional solubility causes near zero HCl mixing ratios. Yes, the temperature dependency is different for other data subsets. But we think that also the ATLAS plots support our argument that an enhanced HCl solubility is not the explanation for the observed discrepancy at 500 K potential temperature. We clarified the argument for selecting 20 June, $\theta$=500 K in the vortex core for this comparison in the revised version.

> **5.** *I am the first to acknowledge that the solubility approach is certainly not "the" explanation or the only cause of the discrepancy and that it has several serious issues. For example, while the running averages (as a function of temperature) agree surprisingly well after application of my 5 K correction (Figs. 1 and 2), individual points show large deviations. To illustrate a particularly bad example, I show data from 15 July 2006 in Figure 3. It is evident that under dehydrified conditions under very cold temperatures, the solubility approach does a very bad job. This is related to the fact that the solubility decreases with decreasing water vapor. However, in the MLS data, very low HCl levels are maintained under dehydrified conditions, excluding the solubility approach as an explanation under dehydrified conditions with the current parameterization. In addition, the 5 K shift is really huge and not compatible with the Luo et al. (1995) parameterization that we probably all use in our models. The solubility parameterization is however not really my area of expertise, and it is difficult to judge for me in how far a different parameterization*

*would be possible and be of help here. Maybe it would be an idea to both introduce a shift in temperature and in water vapor? I think it would be important to discuss the things I mention in the above paragraphs in a little more detail to get a more balanced discussion.*

The temperature dependence in Fig. 7 of the paper did convince us that an underestimation of HCl uptake into the liquid particles is unlikely. However, we must admit that this issue may be not so clear and it would be possible to have a combination of multiple reasons. Therefore we adapted the discussion on HCl solubility and uptake as suggested.

**6.** *I think there are serious issues in the interpretation of your Figure 7 and 8 in section 5. I don't think you can draw the conclusions about the temperature dependency and dependency on sunlight of the HCl discrepancy that you draw here and that this section has to be rewritten. Small changes to the altitude or the date of the figure produce results that can be interpreted completely differently. As an example, I have shown results to compare with your Figure 7 for 46 hPa (my Figure 1) and 31 hPa (my Figure 2) (the 46 hPa and 31 hPa levels are measurement levels of MLS roughly enclosing your 500 K level). These show a completely different temperature dependency for the MLS data for 20.6.2011, which is related to the "tongue" of high HCl visible in Figure 2 of your paper. While the 46 hPa data show a decrease of MLS HCl with decreasing temperature (upper row, second column, Figure 1), the 31 hPa data show an increase. Similar differences in interpretation are found when looking at another date (20.7.2011, right column). Apart from this vulnerability to small changes, I think that this kind of figure is notoriously difficult to interpret, if a) there is more than one process causing the discrepancy, and b) the processes changing HCl have a memory. I.e., the figure is only easy to interpret if HCl is determined instantaneously by a single process. The same issues apply to your Figure 8. I can easily produce a figure, where with increasing sunlight time,*

*the discrepancy does not increase (Figure 4 of this review). But that is even not the main problem with your Figure 8. When I try to reproduce your Figure 8 with the 195 K constraint, I get much lower sunlight hours than you, and no apparent dependency of the discrepancy on sunlight hours below 195 K (my Figure 4, left). When I however just calculate the total sunlight hours (right), the figure looks more similar to your figure (note that I plotted all measurements south of 60 deg S, which explains the higher values for the sunlight hours). I have the impression that there may be a bug in the calculation of the values for your Figure 8, and that the 195 K constraint may be missing. That may also explain why your Figure 7 and 8 look so similar, since temperature and sunlight hours are highly correlated if you skip the 195 K constraint (Figure 5 of this review).*

As outlined above (point 4), we feel that our Figures 7 and 8 are useful to obtain hints on the missing process. That is the case since the time and location are chosen carefully from interpreting the differences between Figs. 2 and 3 of the ACPD manuscript. The comparison between Figure 1 of the review and Fig. 7 of the manuscript is already discussed above (point 4).

With respect to Fig. 8 of the manuscript, we carefully checked again the analysis of the plotted points but we did not find a mistake. Again, we show data at 500 K potential temperature, the data plotted in Fig. 4 of the review 46 hPa have about 3-4 K higher temperatures, thus the time below the 195 K threshold is likely shorter. Also, we should note that the determination of "sunlight hours" is ambiguous. In our study, we used solar zenith angle (SZA) $< 95°$ that includes twilight, which seems different from the definition used in the review. This is mentioned in the revised version.

However, we rewrote the section such that it should be clearer that this comparison only gives hints to a possible solution and is not a proof of a true temperature or daylight dependence.

[Figure]

**7.** *You spend quite some time to show that the process is dependent on sunlight in section 5. But your best explanation (6.3, decomposition of particulate HNO$_3$) is not dependent on sunlight. How does this fit together? This obviously requires some discussion in the paper. Maybe it is related to the issues in section 5.*

The described process indeed would need sunlight. Even though the first step of the proposed mechanism does not include a photolysis reaction, the mechanism itself does include reactions with ClO which are necessary for ClONO$_2$ or HOCl formation. In the cold polar stratosphere, active chlorine at nighttime will be mostly in the form of Cl$_2$O$_2$ instead of ClO. We clarified this point in the revised version.

Specific comments

**8.** *Page 1, line 9: Wouldn't it be better to speak of SD-WACCM (here and else-where) to make clear that you are using the specified dynamics version of WACCM here?*

Yes, changed the manuscript accordingly.

**9.** *Page 1, lines 16-17: Really? You discard this all too easily in my opinion, see major comment.*

In light of the discussion above (4, 5), we will weaken this statement as we cannot exclude increased HCl uptake into the particles. However, we still do not find this to be a likely explanation for the discrepancy.

**10.** *Page 1, lines 20: As long as you don't know the processes that would lead to this HNO$_3$ decomposition, this is no better explanation than the solubility ap-proach.*

To our opinion it is slightly better as it does roughly produce the "fingerprint" of the searched process. Note that we do state clearly that this is a hypothetical process.

**11.** *Page 2, line 30-31: "not yet been reported". You need to change the formulation. In the next sentence, you show that this is not true and that it has been reported in Wegner et al. (and later in Brakebusch et al., Solomon et al., etc.). Maybe just keep the next sentence and delete this one?*

This is correct, see out answer to point 1. We deleted this statement.

**12.** *Sections 2.1-2.3: I would suggest to describe the different heterogeneous schemes (microphysics/chemistry/sedimentation) in some more detail here (as long as it is important for HCl) and to contrast the schemes of the different models to point out in which respect they are different (or identical). That may also be helpful to understand differences in HCl between the models better.*

This is a good idea, even though the schemes are not significantly different with respect to HCl reactions. We will followed this suggestion.

**13.** *Page 7, lines 13-15: I find the tone of this discussion unnecessarily negative. I am certainly not a completely neutral reviewer with respect to my model, but I think a neutral reviewer would have the same comment. First, you write that I did not "find an explanation". But the same is true for this paper. Both of us examined several possibilities for explanations of the discrepancy, and both of us did not find a "final" explanation that would resolve the issue completely. Then, your "best" explanation is as "empirical" as mine. The text here and the following discussion in the next sections however sounds as if my explanation would be empirical and arbitrary, while this is not the case for your "best" explanation. Finally, the addition "but without further evidence why that could be." is completely unnecessary.*
*Please delete this part. I discussed the pros and cons of different explanations on more than two pages, even though I was restricted to keep this short for several reasons. You have very similar problems in your paper ("...the exact mechanism needs to be clarified...") I would suggest a more balanced discussion here, at least that it is mentioned that you have similar problems to the ones that I faced in my publication. A more balanced discussion would also include to mention ATLAS (and the Kuttippurath paper for MIMOSA-CHIM) in the introduction and/or conclusions.*

Yes, we admit that the tone was too one-sided. We think the point is that our paper focuses on the discrepancy, while in the earlier paper it this is not the case. We changed this section accordingly.

**14.** *Page 8, line 5: You write "For this example trajectory". Does that mean that there are other cases which look worse? Or is the trajectory representative?*

This trajectory is meant to be representative being closest to the vortex average ozone and HCl development in the 3-D model. It was the only one that was tested in two chemistry schemes. It did not seem likely that there is a significant difference between the chemistry schemes, however, it seems better to verify this assumption. We will change the text to "For this representative trajectory"

**15.** *Page 9, line 8: It is also worth mentioning that it is also strongly dependent on water vapor.*

This point is now mentioned in the revised version. However, as indicated above, the simulated water vapour is consistent with the MLS observations, at least for the vortex average time series as shown in Tritscher et al. (2018, ACPD)

**16.** *Page 9, lines 12-15: See major comment. I don't think you can draw this conclusion.*

As outlined in points 3c, 4, 5, we think that we can draw this conclusion, however, we will weaken this statement as there could be multiple processes involved.

**17.** *Figure 7: It is interesting that both the CLaMS and the "uncorrected" ATLAS HCl values seem to increase with decreasing temperature. As I said in the major comment, I would not overinterpret a plot like Figure 7, but maybe that hints to something?*

We also would not over-interpret this, but in our opinion this may be caused by the fact that the lowest temperatures are in the core of the polar vortex that also is exposed to the lowest amount of sunlight.

**18.** *Page 9, line 16-24, Figure 8: See major comment. I don't think you can draw this conclusion.*

As explained above (6, 17) we think the conclusion is valid in the sense that this is a hint, not a proof. We re-formulate this accordingly.

**19.** *Page 9, line 29 and Figure 9: I can't follow you here that there is a correlation between dHCl/dt and PSC occurrence based on Figure 9. While there are clear minima in dHCl/dt, the PSC occurrence oscillates somewhere around 0.6 all the time. In fact, there are e.g. higher values for PSC occurrence in end of May than during the periods of low dHCl/dt values later in June and July. It is also not clear to me what the discussion in lines 25-35 is supposed to tell me. I have the impression that this needs to be rephrased.*

[Figure]

It is not so obvious, but still the HCl depletion rate and the PSC occurrence seem to have a relative maximum around 11 June, a minimum around 23 June and again a maximum between 5 and 12 July. In our opinion this is again a hint but clearly not a proof. We clarified this point in this section.

**20.** *Section 6.3: This looks quite promising and very interesting. I would however make more clear somewhere in the paper that there may be other explanations, which possibly work equally well, and which are not discussed in this manuscript. In particular, while your argumentation leading to this explanation is quite logical and sound, the weak point is your assumption of a rather arbitrary fixed rate here, where you don't really know where it comes from. That is quite comparable to the situation with my ad hoc assumption of a 5 K shift in HCl solubility, where I don't really know where it comes from.*

It is true that the suggested process and the rate is speculative which has been clearly stated in the manuscript. As pointed out above, from the analysis of the "fingerprint" of the discrepancy as well as the temperature dependence in the critical region, we think that the process (or the processes) explaining the HCl discrepancy should show a similar fingerprint regarding the HCl depletion rate than our suggested process.

**21.** *Section 6.3: How does the simulated $HNO_3$ compare to MLS $HNO_3$, with the decomposition switched on and off? A better agreement when switching on the decomposition would strengthen your point.*

The additional NAT decomposition results in a small increase of $HNO_3$ in the gas-phase, especially where the $HNO_3$ gradient is large. Figure 4 of this reply shows this comparison. The largest effect of 0.5 ppbv on the 500 K level would be at some spots near the vortex edge around 1 August and also about 0.5 ppbv lower down in the vortex

core at the 400 K around 1 September. In the interesting region of the "HCl tongue" the difference is below 0.15 ppbv. Unfortunately this is below of what could be validated by MLS data due to its vertical resolution and accuracy. Therefore we refrain from including this plot into the revised manuscript.

> **22.** *Page 12, line 10: Is the rate of $10^{-7} s^{-1}$ compatible with the amount of galactic cosmic rays (or secondary electrons with the right energy)? Or is this just a value empirically chosen to get the best fit with the HCl measurements? That should be stated here.*

This is just an empirical value corresponding to a $HNO_3$ decomposition of 0.9% per day chosen at first glance. It corresponds about to the gas-phase $HNO_3$ photolysis rate at 31 hPa and 85° SZA. We do only speculate about the possible missing process and have no proof at all. But this number gives the order of magnitude for such a process. The real solution could likely be more complex, e.g. at least altitude dependent, but we refrain from tuning this process further. The aim of the paper is only to show that such kind of a process would approximately cause the right "finger print".
* * *
**HCl vs current temperature, 20.06.2011, θ=450K**

+ MLS
+ CLaMS

HCl [ppbv]

Temperature [K]

**Fig. 1.** As Figure 7 of the ACPD paper, but for 450 K potential temperature

**HCl vs sunlight hours with T<195K, 20.06.2011, θ=450K**

Legend:
+ MLS
+ CLaMS

(Plot: HCl [ppbv] vs. Sunlight time with T<195K [hours])

**Fig. 2.** As Figure 8 of the ACPD paper, but for 450 K potential temperature.

**Fig. 3.** Comparison of water vapour observed by MLS and simulated by CLaMS in style of Fig. 4 of the ACPD paper. Similar to Fig. 10 of Tritscher et al. (ACPD, 2018).

[Figure]

**Fig. 4.** Comparison of HNO3(g) observed by MLS and simulated by CLaMS in similar style for MLS (top panels), the CLaMS simulations GCR and "NAT decomp" (middle panels) and their differences (bottom panels).

---

## Author Comment (AC2) · 22 May 2018

We thank Reviewer #2 for the constructive review. The review comments are repeated below indented and *in italic letters* followed by our answers.

**Specific comments**

> **1.** *Introduction has to be elaborated with previous modeling studies in the polar stratosphere and HCl comparisons (e.g. Feng et al., Wohltmann et al., and Kuttippurath et al. articles on polar processing and ozone loss studies)*

In the revised version, we mention and discuss also the publications of the other models

that show HCl. For further details, see also our answer to comment 1 of reviewer #1 (Ingo Wohltmann).

> **2.** *As stated in the introduction, the main idea was to check the impact of HCl discrepancy on ozone loss or polar ozone loss chemistry. However, that section is too short and limited to the description of the impact of change in ozone with respect to different model experiments. I would suggest you to calculate the ozone loss (profiles too) and compare with the published results (even for similar winters in the past). This would also give an idea about the model performance in comparison to other models.*

This is a good point. We enhance the discussion of ozone loss by adding the new Figure 14 with a vortex average ozone profile on 1 October when the ozone hole is fully developed. This is done for all three models in comparison with MLS observations. This figure shows that all models are capable of reproducing the ozone hole. It also shows the small effect of the hypothetical simulation "NAT decomp". We also realised that the column ozone depletion for the Arctic winter shown in the old Fig. 14 were based on an earlier model run and were not consistent with the simulations shown in Fig. 4. This was now corrected. However, it does not seem necessary to us to add simulations for other years with the same model setup, as this would be excessive work and more or less a new study.

> **3.** *I think that you missed ClO comparisons in this study, although you have a comparison with ClONO$_2$. You have described a lot about the chlorine partitioning and chemical polar processing (e.g. page 12, line 28—29). Therefore, I think it is important to compare the simulated ClO (from different experiments) with measurements (e.g. from MLS).*

This is correct. The comparison is however somewhat more complicated, as one needs to simulate the diurnal cycle of ClO. For that, we calculated the chemical composition

for the time and location of the MLS data using the chemistry-box-model mode initialised from the CLaMS 3-D simulation on the previous day. Figures 1 and 2 show the comparison on the 500 K level for 20 June and 20 July, respectively. From this comparison, it is evident that in the considered vortex core region ($\Phi_e > 75°$S) the ClO mixing ratios are near zero within the measurement uncertainty. Therefore we cannot gain much knowledge regarding the potentially missing process. Further it is evident, that the hypothetical process included in the simulation "NAT decomp" induces almost no change to the ClO mixing ratios. Nevertheless, we now also mention this aspect regarding the ClO comparison in the revised version of the manuscript.

**4.** *Page 8, Para 2: You stated that the numerical diffusion masks the HCl differences in Eulerian models. However, still the HCl discrepancy is very much apparent in those models/simulations, as demonstrated in this manuscript? So how much is the contribution from numerical diffusion?*

Besides the HCl issue, we think the effect of numerical diffusion itself is an interesting aspect of this study. An exact estimation of the impact of numerical diffusion is however difficult. We show the sensitivity run, in which every 24 h an Eulerian averaging event was triggered in CLaMS. The HCl change in this sensitivity run is on the order of about half of the HCl discrepancy. As the model is set up, it would be technically challenging to do such an Eulerian averaging event at every timestep. A comparison like this would have to be done in a model with otherwise identical setup to quantify the numerical diffusion effect. For the current study, we think that it is enough to point out numerical diffusion as one reason for the inter-model differences without being able to quantify the effect exactly.

**5.** *You have used three different models for this study, which is also the strength of this study. However, a discussion on the ability of synergetic use of the models to be*

*applied for such studies is missing here. Only different test simulations are given. Please include a brief discussion in Section 6, and add few lines in conclusions too.*

Thank you. We followed this suggestion in the revised version.

**Technical**

The typographical and grammatical issues will be updated as suggested.

*Page 5, Line 17: You did not use ClO data?*

We did not use the ClO data for initialisation. The initialisation time is before the period of chlorine activation. Further, it is difficult to derive the partitioning of chlorine from ClO only. However, in the revised version we mention the comparison with MLS ClO as indicated above.

*Page 8, Line 32: Is there any reasons for taking 500 K altitude for this comparison?*

Yes. From Figs. 2 and 3 it is evident that the largest difference between model and data is present in the "tongue" at this level. As here the HCl depletion is still ongoing in the observations, we speculate that this is the location where the possible missing process has the strongest impact. This point will be clarified in the revised version.

*Page 13, Line 9: Numerical diffusion! Then how can we use these models even for this study (e.g. HCl differences)?*

Yes, of course there is a conceptual difference between Lagrangian and Eulerian models. But we think, we can and even should use these different models in this study. While using the Eulerian models for this comparison, we must be aware that the numerical diffusion may cause problems. This is our interpretation for the inter-model

differences. Note that both WACCM and TOMCAT-SLIMCAT are well established used for studying a variety of questions regarding stratospheric ozone.

[Figure]

**Fig. 1.** Comparison of MLS ClO observations with CLaMS for 20.06.2011 at 500K potential temperature pole-ward of 40S: (a) MLS data vs equivalent latitude, (b) corresponding CLaMS results, (c) CLaMS vs MLS.

**Fig. 2.** As Figure 1 but for 20.07.2011.

---

## Author Comment (AC3) · 22 May 2018

We thank Alexander D. James for the comment on our manuscript. Please note our answer below. The comments are repeated indented and *in italic letters* followed by our answers.

> *The authors present a useful summary, and thought-provoking new ideas, relating to a relatively well-known problem in stratospheric chemistry. This is likely to provoke future research and certainly merits publication following some revision.*

We thank for this valuation. Even though that seems to be a relatively well-known problem in stratospheric chemistry, it has not been properly documented in the scien-

tific literature, which is the purpose of this manuscript.

> *The reference to the "dark polar vortex" in the title seems misleading given that the*
> *authors show an apparent correlation of the HCl discrepancy to hours of sunlight.*

This is a valid point. From our Fig. 8, it seems that the missing process needs sunlight, but we are looking at the vortex area with very little sunlight and the observations are mostly in the dark. We therefore changed the title of our manuscript to "On the discrepancy of HCl processing in the core of the wintertime polar vortices"

> *The reviews presented in RC1 and RC2 detail a number of additional processes*
> *which should be discussed in the text. In particular the paper lacks a discussion of*
> *the possible role of solid PSC. The loss of HCl is known to proceed rather differ-*
> *ently on nitric acid hydrate (NAX) or ice surfaces compared to liquid PSCs (Chu et*
> *al., 1993). Hoyle et al. (2013) recently showed a good agreement between CLAMS*
> *and CALIPSO data on PSC type when a heterogeneous nucleation scheme was im-*
> *plemented, leading to a sharp onset in NAX nucleation below 195 K. In James et*
> *al. (2018) we then showed, based a study in our laboratory, that such heteroge-*
> *neous nucleation can be caused by meteoric smoke or fragmented meteoroids, and*
> *can occur at temperatures as high as 197 K for typical stratospheric abundances of*
> *$HNO_3$, $H_2SO_4$ and $H_2O$.*

First of all we should note that we show that the problem of a missing HCl loss occurs in the models, because there is no reaction partner for HCl to react with as both the mixing ratios of $ClONO_2$ and HOCl are close to zero. The main problem is therefore not the available PSC surface area or the PSC type but rather likely a missing process to re-create $ClONO_2$ or HOCl. Therefore the PSC nucleation process is not a central issue of this paper.

The CLaMS simulations do include a parametrisation of NAT and ice nucleation on likely meteoritic particles based on Hoyle et al. (2013) and Engel et al. (2013). The discussion on the PSC nucleation parametrisation has been kept rather short, since it is discussed by Grooß et al. (2014) for NAT and Tritscher et al. (ACPD, 2018) for ice in detail. This manuscript contains the description of the PSC nucleation parametrisation and a detailed comparison of PSC types and distribution of this CLaMS model setup. As the focus of this paper is on the missing HCl depletion, we feel that we should not repeat this documentation here.

*In figure 7 of the current text, the discrepancy is shown not to relate to temperatures below 194 K. I would like to make several points about this approach. Firstly, there is not only a lack of correlation between the model and observations, but in fact the general trend of HCl concentrations is rather different. The model shows a gradual increase in HCl at lower temperatures, whereas the observations show a significant increase below 186 K. This temperature is consistent with the formation of water ice PSC. It would also be informative for the reader if data were included to higher temperatures, to include the formation of possible NAX between 197 and 194 K.*

We cannot completely follow these arguments. We did choose this specific subset of MLS observation points in potential temperature, equivalent latitude and time such that they represent the ongoing HCl depletion to be present both in the model and data. The mentioned increase in the MLS data below 186 K is in fact a "missing decrease" as the observations on this level are unperturbed HCl mixing ratios of about 2 ppbv before. So, if anything, the correlation with temperature alone would tell us about a process not taking place in the presence of ice.

Further, we simply plotted all available data of the chosen subset in the vortex core. There are no warmer observation points present in this specific subset. The idea of this plot was only to exclude the possibility of HCl uptake into liquid (or solid) particles. We do not see the largest discrepancy and the lowest gasphase HCl observations at

the lowest temperatures, which would be the case, if the missing process would be the uptake of HCl into any kind of particles. As also pointed out by the reviewer #1 this kind of figure should only give a hint, but must not be over-interpreted.

*In producing Figure 8 (setting to one side the concerns raised by the reviewers), a cut-off temperature of 195 K is used. This is reasonable if the growth of liquid PSC through deposition of $H_2O$ is the determining factor; however, it is known that significant nucleation of NAX can occur above 195 K (Peter and Grooß, 2012). A more appropriate approach might be to examine the difference between the nucleation scheme used in each model (presumably the same as the Hoyle et al. (2013), Carslaw et al. (2002) and Brakebusch et al. (2013) schemes for CLAMS, SLIM-CAT and WACCM respectively) and the more recent literature. In James et al. (2018) we showed that different parameterisations of nucleation give rather different results in terms of particle concentrations and therefore surface area. A simple and informative experiment would be to correlate the HCl discrepancy to the difference between the onset of nucleation (e.g. time spent between 197 and 195 K for models using a constant volume nucleation rate, see Figure 6 in James et al. (2018)). Since MIPAS data are used extensively another obvious approach would be to relate the HCl (or $ClONO_2$) discrepancy to the data on PSC type which MIPAS provides.*

*It is not possible given the present text to assess whether the presence of solid PSC could explain the discrepancies; however, with the inclusion of the references mentioned above and the subsequent suggestions (see above), this could prove an important process contributing to the HCl discrepancy.*

From our analysis, there seems to be a hint (not a proof) that PSCs may be involved in the potentially missing process. It may well be, that the missing link involves solid PSCs. In fact, we suggest solid NAT in our hypothetical simulation "NAT decomp". However, it is clear from our study, that the standard heterogeneous chemistry on any

of the PSC particles cannot explain the observed HCl depletion, since HCl lacks a reaction partner in the models.

To strengthen our arguments, that CLaMS is able to simulate the basic properties and extent of PSCs, we add also the PSC observation fraction of CLaMS into the lowest panel of Fig. 9.
* * *

---

## Author Response (AR1)

**Comments to the revised version**

We thank the two reviewers and Alexander D. James for their comments that helped us to improve the paper. We addressed the points and changes made to the revised manuscript in detail already in the interactive discussion and do not repeat them here. Also, we did a thorough language editing by the native speaking co-author Martyn Chipperfield. For clarity, we do include here a version of the paper in which the additions and changes to the previous ACPD manuscript have been marked.

Jens-Uwe Grooß.

[revised manuscript text omitted]